# Circadian control of a sex-specific behavior in *Drosophila*

Sabrina Riva[1], Maria Fernanda Ceriani[2], Sebastián Risau-Gusman[1]*,
Diana Lorena Franco[1]*

[1]Medical Physics Department, Bariloche Atomic Center, Comisión Nacional de Energía Atómica (CNEA) and Consejo Nacional de Investigaciones Científicas y Técnicas (CONICET), San Carlos de, Bariloche, Argentina; [2]Laboratorio de Genética del Comportamiento. Fundación Instituto Leloir - IIBBA - CONICET, Buenos Aires, Argentina

**\*For correspondence:**
srisau@cab.cnea.gov.ar (SR-G);
lorena.franco@cab.cnea.gov.ar
(DLF)

**Competing interest:** The authors declare that no competing interests exist.

## eLife Assessment

This **important** study introduces an experimental approach for studying *Drosophila* oviposition rhythms and identifies the subset of circadian clock neurons that mediate the circadian control of oviposition. The authors resolve an inherently noisy rhythm to provide **convincing** evidence by using statistical averaging techniques, which help reduce this noise but at the cost of variation across individual rhythms. This paper will be of interest to anyone interested in insect ovarian physiology, circadian biology, and reproductive fitness.

**Abstract** An endogenous circadian clock controls many of the behavioral traits of *Drosophila melanogaster*. This 'clock' relies on the activity of interconnected clusters of neurons that harbor the clock machinery. The hierarchy among clusters involved in the control of rest-activity cycles has been extensively studied. Sexually dimorphic behaviors, on the other hand, have received less attention. Even though egg-laying, a female characteristic behavior, has been shown to be rhythmic, it remains largely unexplored possibly due to methodological constraints. The current study provides the first steps towards determining the neural substrates underlying the circadian control of egg-laying. We show that, whereas the lateral ventral neurons (LNvs) and the dorsal neurons (DNs) are dispensable, the lateral dorsal neurons (LNds) are necessary for rhythmic egg-laying. Systematically probing the *Drosophila* connectome for contacts between circadian clusters and oviposition-related neurons, we found no evidence of direct connections between LNvs or DNs and neurons recruited during oviposition. Conversely, we did find bidirectional connections between two Cryptochrome (Cry) expressing LNd (Cry + LNds) and oviposition-related neurons. Taken together, these results reveal that Cry + LNd neurons have a leading role in the control of the egg-laying rhythm in *Drosophila* females.

## Introduction

Most organisms are capable of coordinating their physiology and behavior with the 24 hr of day/night cycling generated by the Earth's rotation. These biological rhythms are driven by molecular clocks that are conserved across animals (*Allada et al., 2001*). In *Drosophila melanogaster*, the core clock components comprise a transcription/translation feedback loop with four core proteins: CLOCK (CLK) CYCLE (CYC), PERIOD (PER), and TIMELESS (TIM) (*Hardin, 2005*; *Tataroglu and Emery, 2015*). In brief, CLK and CYC activate transcription of *period* and *timeless* genes, which, once translated into PER and TIM proteins, dimerize and translocate into the nucleus where they bind to CLK and CYC,

thereby inhibiting their own transcription. This molecular feedback loop has a period of approximately 24 hr. In the *Drosophila* brain, this molecular circadian clock is expressed in ~242 neurons, which are organized in different clusters based on gene expression, anatomy, and localization (*Helfrich-Förster, 2005*; *Reinhard et al., 2024*; *Yoshii et al., 2012*). These clusters are: ventrolateral neurons (LNv; encompassing the small and large LNv groups), dorsolateral neurons (LNd), lateral posterior (LPN), and dorsal neurons [DN; separated in DN1, 2, and 3 and further subdivided into anterior (DN1a) and posterior (DN1p) DN1]. Rhythmic locomotor behavior depends primarily on the activity of the sLNvs, LNds, and DN1s, but all clusters contribute to different extents (*Helfrich-Förster, 2005*). Within the lateral neuron group, the sLNv are important because they drive locomotor rhythmicity under free-running (DD) conditions (*Grima et al., 2004*; *Stoleru et al., 2004*), through the release of Pigment Dispersing Factor (PDF), a neuropeptide relevant for communication between clock neurons (*Renn et al., 1999*; *Yoshii et al., 2009*).

Egg-laying is one of the most important female behaviors since it has a profound impact on the fitness of a species. Egg laying is largely governed by successful mating, but is also influenced by circadian and seasonal rhythms and by environmental factors, such as temperature and food availability, among others (*Cury et al., 2019*). The circadian rhythm of oviposition is one of the less studied rhythms in *Drosophila*, possibly due to the challenges involved in monitoring and recording this behavior. The periodic deposition of eggs involves a series of events ranging from the production of oocytes to the choice of the most appropriate substrate for the eggs (*Allemand, 1976*; *Yang et al., 2008*). The circadian nature of this behavior was revealed by its persistence under DD with a period around 24 hr and a peak near night onset (*McCabe and Birley, 1998*; *Sheeba et al., 2001*). Egg-laying rhythmicity is temperature and nutrition-compensated. (*Howlader et al., 2006*). Moreover, oviposition is rhythmic in virgin females as well as in mated ones suggesting that this rhythm is not regulated by the act of mating and, instead, is endogenously driven (*Menon et al., 2014*).

Although oviposition exhibits a circadian component, the molecular and neural substrates that govern this rhythm have only been partially described (*Howlader et al., 2006*). Here, using a semi-automated egg collection device developed in our lab, we examined the contribution of the molecular clock in specific neuronal clusters. By downregulating *per* in subsets of circadian neurons, we determined that even though impairing the molecular clock in the entire clock network reduced the power of the circadian rhythm of oviposition, restricting the disruption to DN1ps or LNvs did not (confirming results obtained by *Howlader et al., 2006* for the sLNv [*Howlader et al., 2006*]). Interestingly, egg-laying rhythms disappeared upon targeting a subset of LNd neurons (Cry + LNd) through RNAi-mediated *per* downregulation, a condition where rhythmicity of locomotor activity patterns remained unaltered, suggesting a leading and very specific role of LNd clocks/neurons in the control of oviposition. Finally, the assessment of the synaptic connectivity between clock and oviposition controlling neurons using the Janelia hemibrain dataset revealed direct synaptic connections between this subset of LNds and oviposition neurons, which is consistent with the essential role of LNds neurons in the control of this behavior.

## Results

### Egg-laying is rhythmic when registered with a semi-automated egg collection device

Oviposition in *Drosophila* is one of the less studied behaviors regulated by the circadian clock. This is probably due to the difficulties involved in monitoring and recording it, and the current lack of standard devices to accomplish this. Egg collection and counting is usually done manually, making the experiments particularly demanding and labor-intensive. In this approach, eggs are typically collected every 4 hr (sometimes also every 2 hr), which usually implies transferring the fly to a new vial or extracting the food with the eggs and replacing it with fresh food in the same vial (*Menon et al., 2014*). Either way, this implies disturbing the fly several times a day, which could alter the normal rhythm of oviposition, and demands the intervention of an experimenter every 4 hr during several days. In order to avoid this, we developed a semi-automated egg collection device where 21 flies are individually housed, and each enclosure is slowly shifted every 4 hr from one food patch to a new food patch. Once per day, all food patches are collected and eggs are manually counted (see **Material and methods;** *Figure 1—figure supplement 1* **and** *Figure 1—video 1* for more details).

The egg records of fruit flies display several distinct features that complicate the assessment of rhythmicity. The recorded variable is discrete and typically takes only a few values, depending on the sampling frequency. The more often it is sampled, the fewer number of values the variable takes. Thus, it is only possible to sample it a few times per cycle (usually less than ten). For noisy signals, this feature makes the assessment of rhythmicity much more complicated, compared with signals that have similar noise, but are sampled more frequently (as is the case for activity records). The particularities of egg-laying in *Drosophila* add further complications. First, the number of eggs laid shows a clear decreasing trend over time. Second, this behavior appears to be particularly 'noisy,' in the sense that the decision to lay an egg depends not only on the time of the day, but also on many other stimuli (*Cury et al., 2019*). As a consequence, any method that tries to assess the presence of a rhythm must be able to subtract, at least partially, the effects of trend and noise.

In order to achieve this, we have considered the register of the eggs laid by a 'rhythmic' female as a periodic signal corrupted by random noise (whereas for arrhythmic females, we assume the record is completely random). It is well known that light is a powerful zeitgeber for activity, synchronizing all the flies in a population to the light-dark (LD) cycle, and that even after lights off, activity remains synchronized between flies for quite a long time. Since this is caused by the synchronization of cellular clocks, we assume that a similar effect occurs for oviposition. Thus, under LD conditions, or in the first days of DD, it can be assumed that the periodic signal is the same for all flies, whereas the noisy component is different for each individual. In this case, it is well known (see e.g. *van Drongelen, 2007*) that averaging of several 'replicates' (individual flies, in our case) leads to an improvement of the signal-to-noise ratio allowing the emergence of the underlying rhythm. As this approach seems to be better suited for detecting noisy rhythms, in the rest of the paper the assessment of rhythmicity will be based on the record of eggs laid during a given period of time averaged over all females of each genotype. *Figure 1* shows that, when eggs are collected with our device in LD, *Canton-S* females have a peak of egg deposition at the end of the day, as previously described in the literature (*Bailly et al., 2023*; *Howlader et al., 2006*; *Sheeba et al., 2001*; *Figure 1A*). *Figure 1E* shows that only about 70% of individuals are rhythmic (with a period close to 24 hr *Figure 1F*), although the average egg-laying profile is robust with a period around 24 hr (*Figure 1B*). Under DD conditions, the percentages of rhythmic individuals are the same as in LD (*Figure 1E*) and the average profile also shows a robust rhythm with a period of 24 hr (*Figure 1D*). It is well known that in locomotor activity experiments more than 90% of the flies are rhythmic in DD, and virtually all flies are rhythmic in LD (see e.g. *Deppisch et al., 2022*; *Shindey et al., 2017*). This shows that, at the individual level, egg-laying is much less robust than locomotor activity, compounding the difficulty of observing the influence of the circadian clock on this behavior.

In order to check that our egg collection device does not in any way entrain egg laying in flies, we tested it with perS mutants (*Konopka and Benzer, 1971*). These flies are known to display a short period of locomotor activity, both in males (*Konopka and Benzer, 1971*) and in females (*McCabe and Birley, 1998*). Additionally, it has been shown that oviposition in perS flies is rhythmic, with a period of 20.6+/-0.9 hr (*McCabe and Birley, 1998*). Our results for this mutant are in agreement with this (*Figure 1J* shows a period of the average at 18.8+/-1.1 hr), thus confirming that flies are not entrained by our egg collection device. It is important to notice that in the *yellow white* (*YW*) control strain, the individual rhythmicity was rather low (around 50% *Figure 1K*), but the average is strongly rhythmic (*Figure 1H*), thus confirming the need for an analysis method that can take into account the low signal-to-noise-ratio of this behavior.

## Downregulation of *per* in clock neurons causes a dramatic reduction in the power of egg-laying rhythms

The circadian system of *Drosophila* comprises a central clock located in the brain that controls locomotor behavior, as well as peripheral clocks located in many tissues, that regulate distinct behaviors (*Franco et al., 2018*; *Ito and Tomioka, 2016*). For example, olfaction rhythms are controlled by peripheral clocks in the antennae, and eclosion rhythms are partially controlled by peripheral clocks in the prothoracic gland (*Myers et al., 2003*; *Tanoue et al., 2004*). Earlier work proposed that oviposition might also be controlled by a peripheral clock (*Manjunatha et al., 2008*). In order to test for this possibility, we used RNA interference (RNAi) to knock down the clock protein PER (*Herrero et al., 2017*; *Zhang et al., 2021*) in all clock neurons in the brain (using the *Clk*856 driver [*Gummadova*

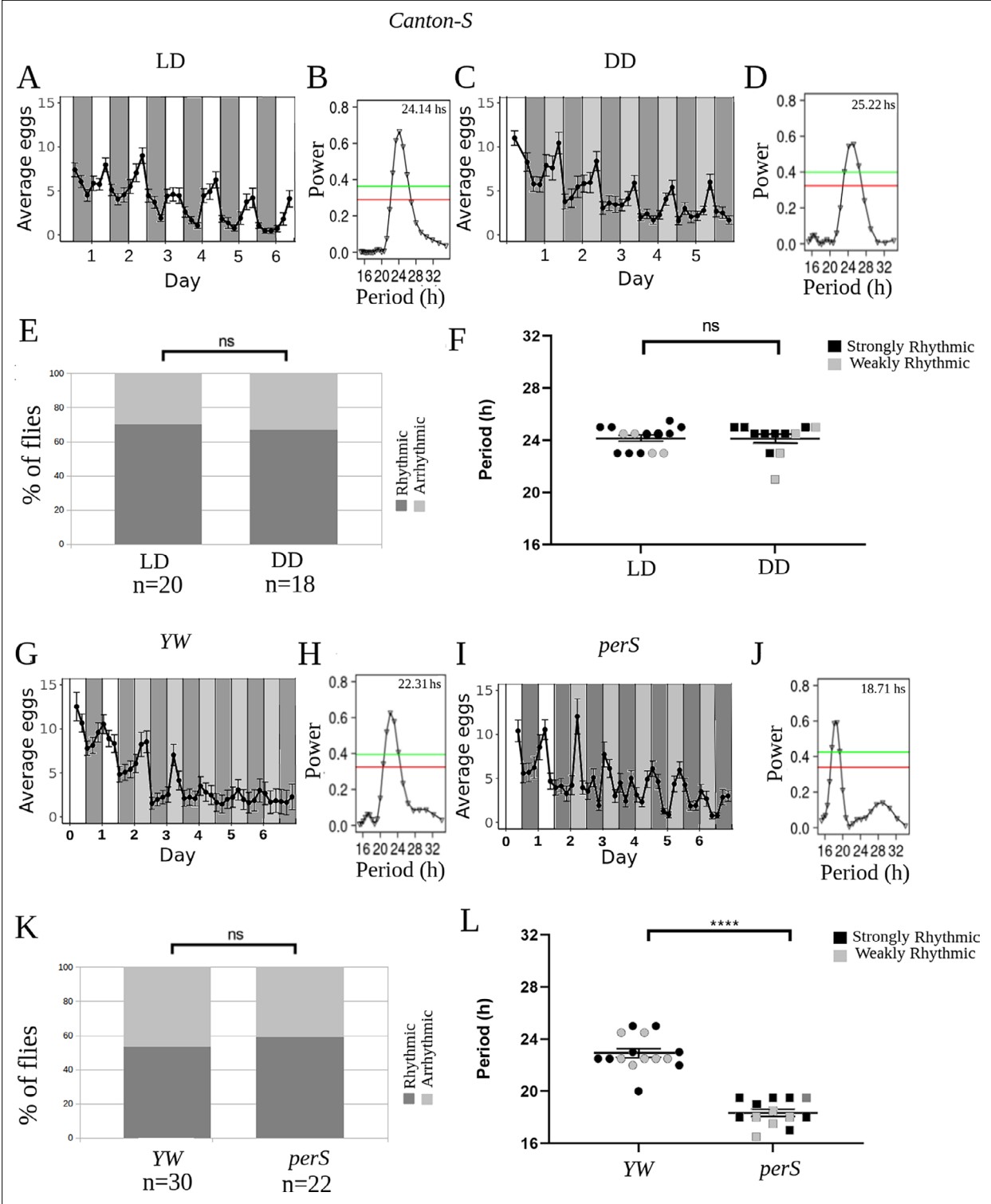

**Figure 1.** Oviposition in *Drosophila* is rhythmic when registered with our semiautomated egg collection device. (**A, C, G, I**) Average eggs collected as a function of time in four different experiments (*Canton-S, LD*: A, Canton-S, DD:C; *yellow white* (*YW*): G; *perS*: I). The average was made with all flies (Strongly rhythmic, weakly rhythmic, and arrhythmic). White and dark gray bars represent periods of lights on and off, respectively (light-dark, LD), whereas light gray bars represent subjective days, DD, (i.e. times where lights were on at rearing, but are now off). (**B, D, H, J**) Lomb-Scargle periodograms of all genotypes, made with all flies, strongly rhythmic, weakly rhythmic and arrhythmic. Red and green horizontal lines represent significances of 0.05 and 0.01, respectively. (**E, K**) Percentage of females with rhythmic oviposition (E: *Canton-S* in LD and DD, K: *YW* and p*erS*). The

*Figure 1 continued on next page*

*Figure 1 continued*

rhythmic flies include both strongly and weakly rhythmic flies. (**F, L**) Period of oviposition rhythms for rhythmic individual flies (F: *Canton-S* in LD and DD, p=0.82; L: *YW* and *perS*, p<0.0001, Cohen´s d=3.84). ns: non-significant, \*\*\*p<0.001 (chi-squared test for scatter plots and proportions).

The online version of this article includes the following video and figure supplement(s) for figure 1:

**Figure supplement 1.** Semi automated egg collection device.

**Figure 1—video 1.** Demonstration of the functioning of the semi automated egg collection device.

https://elifesciences.org/articles/103359/figures#fig1video1

*et al., 2009*]), thereby disrupting their molecular clock. First, we tested the effectiveness of this tool by monitoring the locomotor activity of males with their endogenous clock silenced in all clock neurons (*Clk*856-Gal4>UAS-*per^{RNAi}*). *Supplementary file 1A* shows that these flies exhibited a significant decrease in locomotor rhythmicity compared to their genetic controls. Having confirmed the effectiveness of the RNA interference, we then monitored egg-laying in females from the previously mentioned lines.

As shown in *Figure 2D and E*, when PER is knocked down in all clock neurons, oviposition rhythmicity is drastically reduced compared to controls *Figure 2B and C*. Notice, however, that the peak observed in *Figure 2C* has not disappeared completely in 2E, and is in fact not far from the significance threshold. This could be due to some remaining rhythmicity in some individuals (although noise precludes the observation of this at the individual level). Interestingly, when locomotor activity is studied in male of this genotype, there is a fraction (less than 40%) of individuals that remain rhythmic in DD (see *Supplementary file 1A*).

This could be due either to the contribution of some peripheral oscillator/s relevant for the control of this behavior, or the presence of some residual PER in clock cells (*Herrero et al., 2017*).

## LNv and DN1 neurons are not necessary for egg-laying rhythmicity

We next sought to establish the role of the most important circadian groups in the control of oviposition. We concentrated on three groups: LNv, DN1p, and LNd neurons. The LNvs are known to be essential for the maintenance of circadian locomotor activity rhythms, and are the only ones that express the PDF neuropeptide (*Renn et al., 1999*; *Yoshii et al., 2009*). However, we found that *per^{RNAi}* mediated disruption of the molecular clock in PDF + cells does not abolish the time-of-day-dependent oviposition (see *Figure 2G, H, I and J*). This is in line with previous results describing rhythmic oviposition after ablation of all PDF + neurons (*Howlader et al., 2006*). In addition, we observed a shortening of the egg-laying period when compared to controls, (20.74 hr+/-1.21 hr vs 24.35 h+/-1.83 hr, respectively, *Figure 2H and J*). This shortening can also be observed at the individual level (*Figure 2—figure supplement 1*). Taken together, these results suggest that these neurons may have some influence on the control of this behavior.

The DN1 neurons are not essential for the maintenance of locomotor rhythms in DD, but they contribute to the siesta (*Guo et al., 2016*) and generation of evening activity (*Helfrich-Förster et al., 2007*; *Zhang et al., 2010*) when the peak of egg deposition occurs. Interestingly, it has recently been shown that a posterior subset (DN1p) drives the circadian rhythm of oogenesis through the neuropeptide allatostatin C (*Zhang et al., 2021*). To examine whether the molecular clock of DN1p neurons is involved in the control of rhythmic oviposition, we evaluated the impact of PER downregulation on mated females expressing the *Clk*4.1 driver (which is expressed in ~10 DN1ps per hemisphere *Zhang et al., 2010*). As *Figure 2L, M, N and O* shows, the disruption of the clock in these neurons does not alter the rhythmicity of egg-laying in DD. To further confirm these results, and in the process examine if some other features of the DN1p could contribute to the control of oviposition, we silenced these neurons through the expression of the inward rectifier potassium channel *kir*2.1 (*Baines et al., 2001*). Silencing DN1p neurons did not affect the rhythmicity of egg-laying in females which showed similar patterns to those exhibited by control females (*Figure 2Q, R, S and T*). Taken together, these results show that neither LNv nor DN1p neurons play an important role for rhythmic oviposition.

## The molecular clock in E neurons is necessary for rhythmic egg-laying

The LNd neurons include 6 cells whose expression pattern is very heterogeneous (*Hermann-Luibl and Helfrich-Förster, 2015*; *Ma et al., 2021*). This group contains a subset of 3 Cry + LNd neurons that

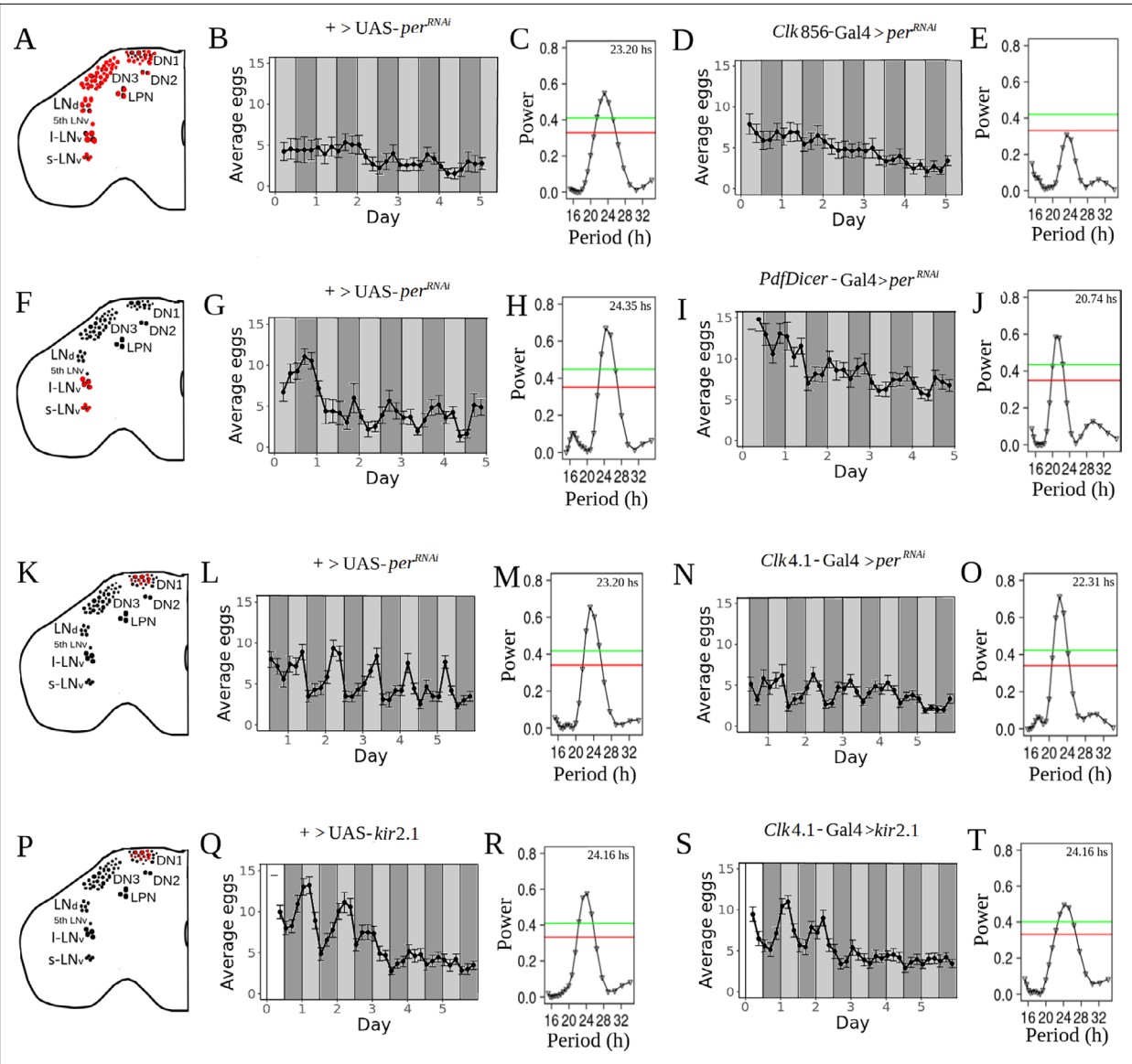

**Figure 2.** Circadian rhythmicity of oviposition is dramatically reduced when the molecular clock is disrupted in all clock neurons, but not when only LNv or DN1 neurons are affected. (**A, F, K, P**) Schematic diagram of the neurons (painted in red) where the molecular clock has been disrupted. (**B, D, G, I, L, N, Q, S**) Average eggs collected as a function of time. The average was made with all flies (strongly rhythmic, weakly rhythmic and arrhythmic). White and dark gray bars represent periods of lights on and off, respectively (light-dark, LD), whereas light gray bars represent subjective days, DD, (i.e. times where lights were on at rearing, but are now off). (**C, E, H, J, M, O, R, T**) Lomb-Scargle periodograms of all genotypes (made with all flies, rhythmic, weakly rhythmic and arrhythmic). Red and green horizontal lines represent significances of 0.05 and 0.01, respectively. (**B, C**) :+>UAS-*per*$^{RNAi}$, n=22. (**D, E**) *Clk*856-Gal4>UAS-*per*$^{RNAi}$, n=26. (**G, H**) +>UAS-*per*$^{RNAi}$, n=18. (**I, J**) *PdfDicer*-Gal4 >UAS-*per*$^{RNAi}$, n=34. (**L, M**) +>UAS-*per*$^{RNAi}$, n=38. (**N, O**) *Clk*4.1-Gal4>UAS-*per*$^{RNAi}$, n=40. (**Q, R**) +>UAS-*kir*2.1, n=36. (**S, T**) *Clk*4.1-Gal4>UAS-*kir*2.1, n=35.

The online version of this article includes the following figure supplement(s) for figure 2:

**Figure supplement 1.** Disruption of the molecular clock in PDF+ neurons causes a shortening of the egg-laying period at the individual level.

express both *cry* (which encodes the light-sensing protein cryptochrome) and *pdfr* (the PDF receptor), and 3 Cry- LNd neurons that express neither of these genes (*Im et al., 2011*; *Yoshii et al., 2008*). Even though the LNds are not essential for rhythmic locomotor activity under constant conditions, the Cry + LNds, together with the fifth sLNv, termed the 'E cells,' are the main drivers of evening activity (*Stoleru et al., 2004*). These two LNd groups greatly differ in their connectivity to the rest of the brain, since Cry + LNds have both many more outputs and inputs than the Cry- LNds (*Shafer et al., 2022*). In order to assess the role of the E neurons in the control of oviposition, we employed a Gal4 driver,

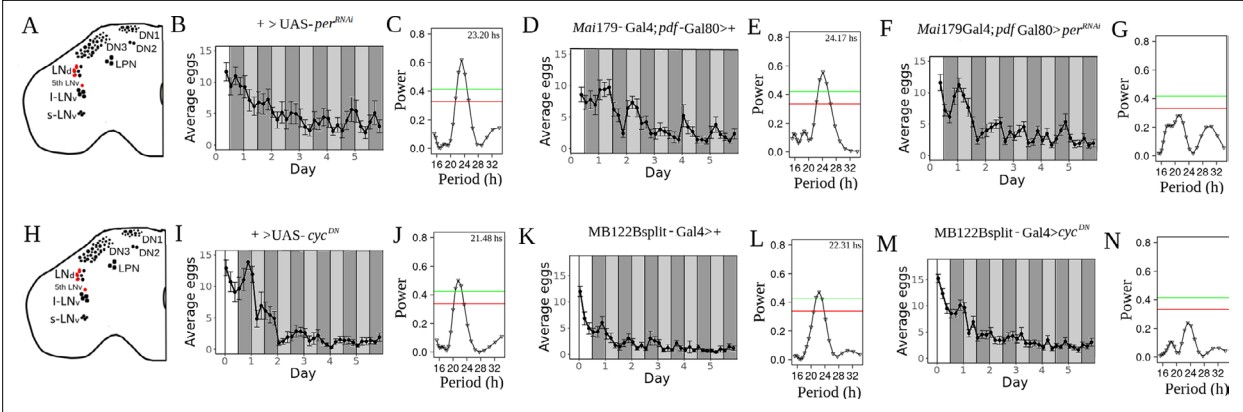

**Figure 3.** Disruption of the molecular clock in E neurons drastically reduces the circadian rhythmicity of oviposition. (**A, H**) Schematic diagram of the neurons (painted in red) where the molecular clocks have been disrupted. (**B, D, F, I, K, M**) Average eggs collected as a function of time. The average was made with all flies (Strongly rhythmic, weakly rhythmic and arrhythmic). White and dark gray bars represent periods of lights on and off, respectively (light-dark, LD), whereas light gray bars represent subjective days, DD, (i.e. times where lights were on at rearing, but are now off). (**C, E, G, J, L, N**) Lomb-Scargle periodograms of all genotypes (made with all flies, rhythmic, weakly rhythmic and arrhythmic). Red and green horizontal lines represent significances of 0.05 and 0.01, respectively. (**B, C**) +>UAS-per$^{RNAi}$, n=15. (**D, E**) *Mai*179-Gal4; *pdf*-Gal80>+, n=17. (**F, G**) *Mai*179-Gal4; *pdf*-Gal80 >UAS-per$^{RNAi}$, n=28. (**I, J**) +>UAS-cyc$^{DN}$, n=15. (**K, L**) MB122Bsplit-Gal4>+, n=14. (**M, N**) MB122Bsplit-Gal4>UAS-cyc$^{DN}$, n=30.

The online version of this article includes the following figure supplement(s) for figure 3:

**Figure supplement 1.** Disruption of the molecular clock in Cry+ lateral dorsal neuron (LNd) neurons drastically reduces the circadian rhythmicity of oviposition.

**Figure supplement 2.** Individual periodograms for the records obtained after applying detrending for the experiment in DD with Mai179-Gal4; pdf-Gal80 >UAS-per RNAi (n=28) female flies described in the main text.

**Figure supplement 3.** Individual periodograms for the records obtained after applying detrending for the experiment in DD with Mai179-Gal4; pdf-Gal80>+ (n=17) female flies described in the main text.

**Figure supplement 4.** Individual periodograms for the records obtained after applying detrending for the experiment in DD with +>UAS-per RNAi , (n=15) female flies described in the main text.

**Figure supplement 5.** Individual periodograms for the records obtained after applying detrending for the experiment in DD MB122B split-Gal4>UAS-cyc DN (n=30) female flies described in the main text.

**Figure supplement 6.** Individual periodograms for the records obtained after applying detrending for the experiment in DD with MB122B split-Gal4>+ (n=14) female flies described in the main text.

**Figure supplement 7.** Individual periodograms for the records obtained after applying detrending for the experiment in DD with +>UAS cyc DN (n=15) female flies described in the main text.

*Mai*179-Gal4; *pdf*-Gal80 (*Grima et al., 2004*; *Picot et al., 2007*). This driver is expressed in the fifth sLNv, 3 Cry + LNds, and also drives weak expression in a small subset of DN1s. As *Figure 3G* shows, impairing clock function in E neurons results in a drastic reduction of rhythmic egg-laying compared with the genetic controls (*Figure 3C and E*). To further confirm this observation, we used another approach to disrupt the molecular clock in those neurons, resorting to the very specific MB122B driver which is expressed in three Cry + LNds and 5-th sLNv neurons (*Duhart et al., 2020*; *Guo et al., 2017*). We also used a different UAS sequence, a dominant negative version of the clock gene *cycle* (*Krishnan et al., 1999*). As expected, under these conditions, rhythmic egg laying was also severely compromised (*Figure 3M and N*), as compared with the genetic controls (*Figure 3I, J, K and L*). Moreover, when we used RNAi to knock down the clock protein PER with the MB122B driver, the results were very similar to those obtained with the Mai179-Gal4; pdf-Gal80 driver (*Figure 3—figure supplement 1*). Thus, we conclude that the molecular clock in E neurons is necessary for the generation of egg-laying rhythms.

The appearance of two non-significant peaks in *Figure 3G* (and *Figure 3—figure supplement 1*) might suggest that these genetic manipulations of the LNd neurons generate weakened complex rhythms (*Beckwith and Ceriani, 2015*) instead of complete arrhythmicity. This would mean that, at the individual level, there should be either many flies with a corresponding complex rhythm, or two weakly rhythmic subpopulations, each one accounting for one of the peaks of the average periodogram.

However, an examination of the individual periodograms (*Figure 3—figure supplements 2–7*) does not provide convincing evidence to support any of these two hypotheses. On the other hand, when the method is applied to synthetic, completely random time series, it can generate periodograms with two non-significant peaks (as shown in panels **2, 3, and 4 of** *Appendix 1—figure 12*). Thus, we favor the hypothesis that the two peaks present in *Figure 3G* (*Figure 3—figure supplement 1*) are spurious.

It has recently been shown that disrupting the molecular clock in E cells by means of the MB122B driver did not alter locomotor activity rhythms (*Bulthuis et al., 2019*). This would imply that locomotor activity and oviposition rhythms are controlled by distinct groups of clock neurons. However, *Bulthuis et al., 2019* only assessed the locomotor activity of male flies, and it is well known that, unlike males, females undergo important changes in their activity after mating, such as the loss of the midday siesta (*Isaac et al., 2010*) or the morning anticipation (*Riva et al., 2022*). For these reasons, we decided to examine the influence of the molecular clock of E cells directly on the locomotor activity of mated females. The standard setup for assessing locomotor activity is the *Drosophila* Activity Monitoring (DAM) system (*Pfeiffenberger et al., 2010*), where flies are housed in small glass tubes (of 5 mm in diameter). Such arena does not offer an adequate environment for testing flies that lay eggs during several days; instead, we monitored activity using an alternative system developed in our lab, where flies are housed in much larger cubicles and their activity is recorded by a video system (see Methods and [*Riva et al., 2022*]). Interestingly, mated females with disrupted E cell clocks were as rhythmic as control flies (*Figure 4*), showing that there is no sex-specific contribution of this cluster to the control of rhythmic rest-activity cycles, which opens the possibility that E cells differently contribute to the control of oviposition and locomotor activity. In other words, our results show that these two important rhythmic behaviors are primarily controlled by distinct groups of clock neurons.

## Two Cry+ LNd neurons directly contact oviIN

If the circadian clock influences oviposition, at least some clock neurons should be connected, directly or indirectly, to the neurons in the brain that control the motor program leading to egg deposition. These neurons, that were recently characterized (*Vijayan et al., 2023*; *Wang et al., 2020*; *Zhou et al., 2014*), include: the oviposition descending neurons (oviDN, 5 per hemisphere), the oviposition excitatory neurons (oviEN, 1 per hemisphere), the oviposition inhibitory neurons (oviIN, 1 per hemisphere), and pC1 neurons (5 per hemisphere). The best tool in hand to systematically probe the connectivity between circadian clock and oviposition-related neurons is the *Drosophila* hemibrain connectome (*Scheffer et al., 2020*). This is the result of the FlyEM connectomics project, which used electron microscopy images of the brain of a 5-day-old female fly to reconstruct the connectivity map (or connectome) of ~25,000 neurons, mostly from the central part and the right lateral part of the brain.

28 of these neurons have been identified as clock neurons, and are located in the right part of the brain (except for one LPN [*Scheffer et al., 2020*]). These include the best characterized circadian groups (sLNvs, lLNvs, LNds). For other groups, such as the DN1s and LPNs, only a minor fraction have been identified, whereas for groups DN2 and DN3, no neurons have yet been identified in this volume.

Interestingly, almost all oviposition-related neurons have been identified in the hemibrain (only two oviDN on the left hemisphere are still missing), and very recently, a couple of additional groups called U and G, comprising 2 and 5 neurons each, respectively, have also been shown to be important for oviposition and mapped to the connectome (*Vijayan et al., 2023*). This makes a total of 25 oviposition-related neurons identified thus far.

To address the possibility of direct connections between clock neurons and neurons involved in egg laying, we probed the hemibrain connectome for these connections. The full results are displayed in *Supplementary file 1B and C*, but in short, we found that whereas there are no connections between LNv or DN1 neurons and oviposition neurons, LNds contact oviIN and pC1 neurons (*Figure 5*), further supporting the experimental findings described in previous sections. Interestingly, different subgroups of E cells (*Shafer et al., 2022*) show distinct patterns of connectivity to oviposition neurons: E3 neurons (comprising Cry- LNd1, LNd2, and LNd3) only contact the pC1 group, while E1 neurons (comprising Cry +LNd4 and LNd5) only contact the oviIN neurons (*Figure 5*). On the other hand, the E2 neurons (comprising Cry + LNd6 and fifth sLNv) make no contacts with oviposition-related neurons.

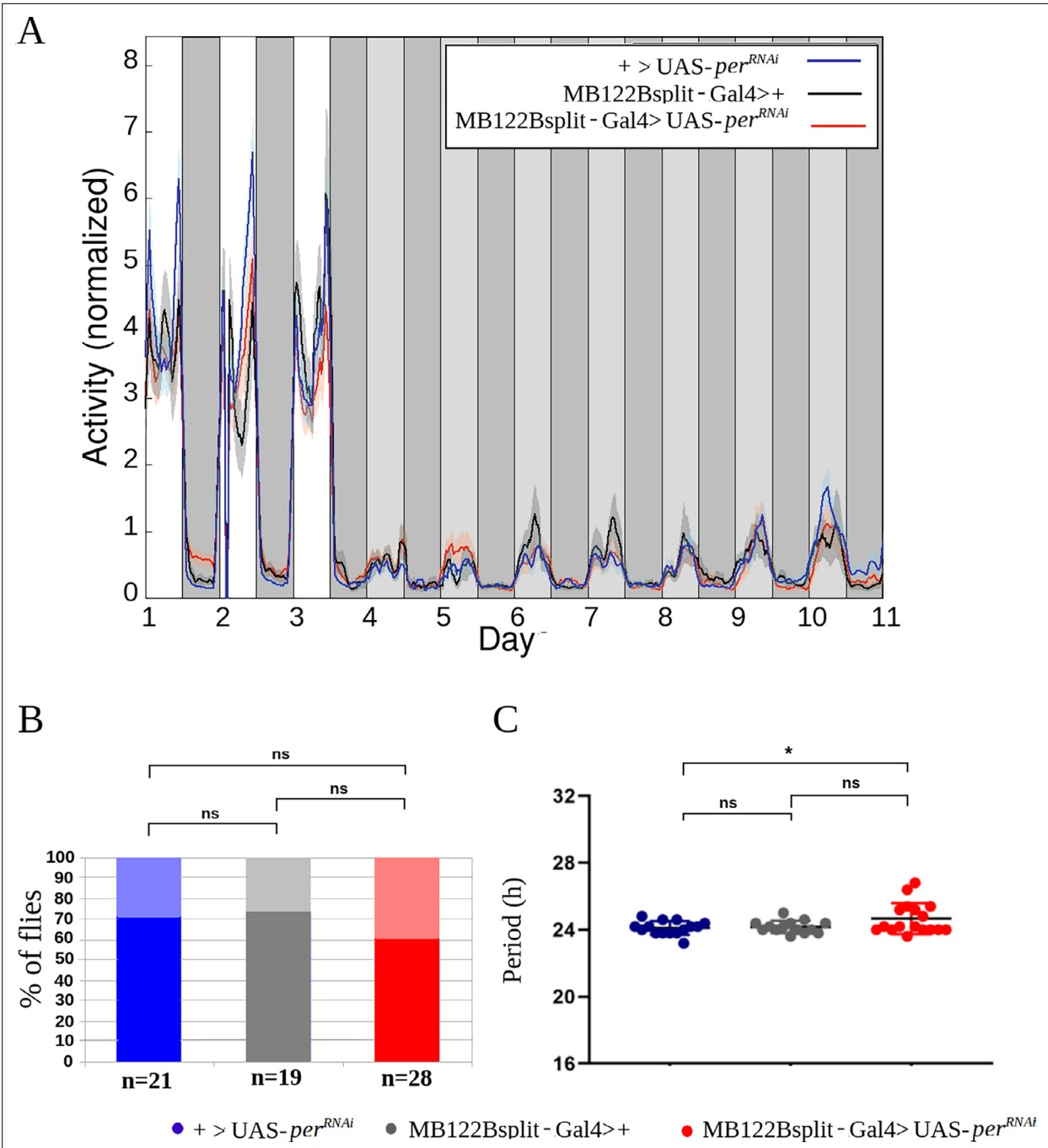

**Figure 4.** Disruption of the molecular clock in E neurons does not alter the circadian rhythmicity of locomotor activity of mated female flies. (**A**) Average activity recording during 3 days in light-dark (LD) and 7 in DD. Light gray bars represent subjective days. (**B**) Percent of rhythmic flies in DD (+>UAS-per$^{RNAi}$ vs MB122Bsplit-Gal4>+p = 0.87;+>UAS-per$^{RNAi}$ vs MB122Bsplit-Gal4>UAS-per$^{RNAi}$ p=0.44; MB122Bsplit-Gal4>+vs MB122Bsplit-Gal4>UAS-per$^{RNAi}$ p=0.36). (**C**): Periods of locomotor activity in DD of individually rhythmic flies (+>UAS-per$^{RNAi}$ vs MB122Bsplit-Gal4>+p = 0.96;+>UAS-per$^{RNAi}$ vs MB122Bsplit-Gal4>UAS-per$^{RNAi}$ p=0.044, Cohen's d=0.77; MB122Bsplit-Gal4>+vs MB122Bsplit-Gal4>UAS-per$^{RNAi}$ p=0.091). Mean ± SEM indicated. Each dot represents one fly. ns: non-significant, *$p<0.05$ chi-squared test for the comparison of proportions, and one-way ANOVA with post-hoc Tukey for the scatter plots.

This evidence suggests that the subset of E cells responsible for the loss of rhythmic oviposition are likely E1 neurons.

Interestingly, we noticed that connections between clock and oviposition neurons are bidirectional, although the E1-oviIN connection is stronger in the direction from the clock to oviposition neurons.

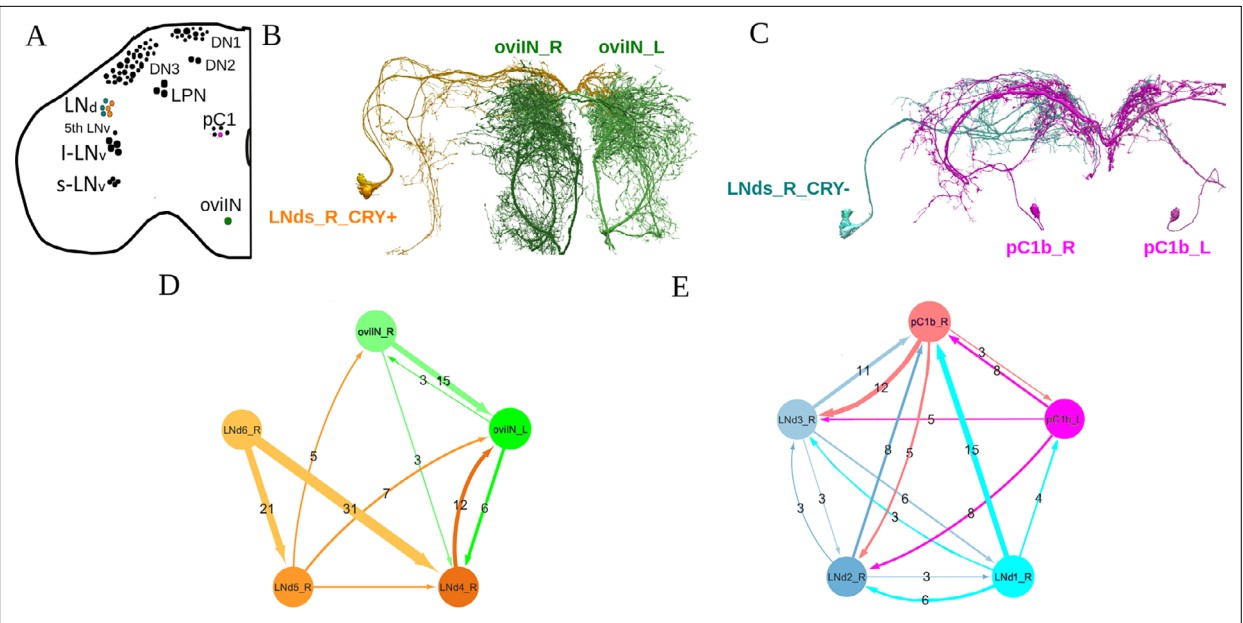

**Figure 5.** Direct synaptic connections between circadian clock neurons and oviposition-related neurons in the hemibrain dataset. (**A**) Schematic diagrams showing the different neuron clusters analyzed (in color). (**B**) Connection between Cry + lateral dorsal neuron (LNd) and oviIN neurons. (**C**) Connection between Cry- LNd 2 and pC1b neurons. (**D**) Network representation of the connectivity between Cry + LNd and oviIN neurons. (**E**) Network representation of the connectivity between Cry- LNd and pC1b neurons. Numbers give the number of synaptic contacts, which represent the strength of the connections. Only intermediate (between 3 and 9 synaptic contacts) and strong (>9 synaptic contacts) connections are considered.

The online version of this article includes the following figure supplement(s) for figure 5:

**Figure supplement 1.** Direct synaptic connections between lateral posterior neuron (LPN) and oviposition-related neurons in the hemibrain dataset.

This analysis suggests that, even though some feedback is anticipated, the E1 neurons are likely the ones providing temporal information to the oviposition circuit.

Additionally, aside from the circadian groups considered so far, we found strong, bidirectional connections between the clock neurons LPN (*Reinhard et al., 2022*) and the oviIN, with the strongest connections going from oviIN to LPN (*Figure 5—figure supplement 1* and *Supplementary file 1B and C*).

## Discussion

The oviposition rhythm is one of the less studied behavioral traits of *Drosophila*. One possible reason is merely technical, since egg collection and counting are very laborious processes and there are no standard devices (just as the DAM system for locomotor activity) to reproducibly evaluate it. Data analysis and rhythmicity assessments are also particularly difficult because the presence or absence of a rhythm must be established with only approximately 6 data points per cycle (in contrast to locomotor activity time series, where hundreds of data points per cycle are obtained), which can take only a few discrete values (typically, from 0 to 10). The analysis is further complicated by the fact that the daily average of eggs laid decreases with time (*Kaufman and Demerec, 1942*). To tackle these limitations, we developed a semi-automated method for egg collection and a new pipeline for data analysis. We observed that the oviposition rhythm is less consolidated than locomotor activity rhythms. Surprisingly, this happens even in LD, where time-of-day cues are provided by light, the most important *zeitgeber* for diurnal animals. Egg-laying is even less rhythmic in DD since the information from the endogenous clock is one among many internal and external factors that influence the decision of laying an egg. In a sense, the oviposition record of a single fly can be considered as a periodic signal with strong noise added to it. Such data might not even look rhythmic when analyzed individually. However, if we assume that the periodic component is the same for all flies in a population, and the noise is different for each individual, the periodic component can be extracted by averaging

the oviposition data. This led us to prefer periodograms of averaged data for the assessment of the rhythmicity of a given genotype.

Even though there have been some advances in the understanding of the relationship between circadian clock and oviposition (*Howlader and Sharma, 2006*; *Manjunatha et al., 2008*), there is still no information about the relative importance of the different neuronal groups in driving this behavior. Here, we downregulated the molecular clock in subsets of circadian neurons in order to establish their role in the control of circadian oviposition. First, we found that animals with downregulated *per* expression in all clock neurons of the brain display a drastic reduction of rhythmic oviposition, underscoring a key role of the central clock (as compared with peripheral clocks) in the control of the timing of oviposition.

Even though we also disrupted the molecular clock in PDF-expressing neurons as well as in the DN1p group, egg-laying rhythms were not abolished, in agreement with previous reports (*Howlader et al., 2006*). We cannot rule out a contribution of PDF-expressing neurons, since we observed a shortening of the oviposition period. Interestingly, a previous work (*Howlader et al., 2006*) reported a similar period shortening in *pdf01* mutant flies, but not when PDF-expressing neurons were ablated.

Taken together, these results show that communication between either group and any other neurons is not necessary for rhythmic oviposition. This is in stark contrast with the situation for locomotor activity, which has been shown to become arrhythmic when PDF + cells are ablated (*Grima et al., 2004*; *Stoleru et al., 2004*). Interestingly, however, it has been recently shown that disrupting the molecular clock in these same neurons does not alter locomotor rhythmicity (*Delventhal et al., 2019*; *Schlichting et al., 2019*) suggesting a more complex scenario.

Since it has been shown that the DN1p generate the rhythm of oogenesis (*Zhang et al., 2021*), one might wonder why the disruption of their molecular clock or their electrical silencing does not also abolish the rhythm of oviposition. It is important to understand that the suppression of the oogenesis rhythm does not necessarily have an impact on rhythmic egg laying. The persistence of oviposition rhythm suggests that it does not depend on the rhythmic production of oocytes, but is instead an intrinsic property of the motor program which is the hypothesis more supported by our results.

Somewhat surprisingly, the disruption of the molecular clock in E neurons led to an almost complete loss of the oviposition rhythm. In contrast, downregulating the clock in these neurons did not affect rhythmic locomotor activity in mated females, underscoring a degree of specificity. Interestingly, in males, the Cry + LNds have been singled out as responsible for driving the activity peak at dusk (evening peak, [*Guo et al., 2017*]), which is the time when, in females, egg laying has a peak. As far as we know, oviposition is the first female-specific behaviour specifically controlled by the LNds. In males,

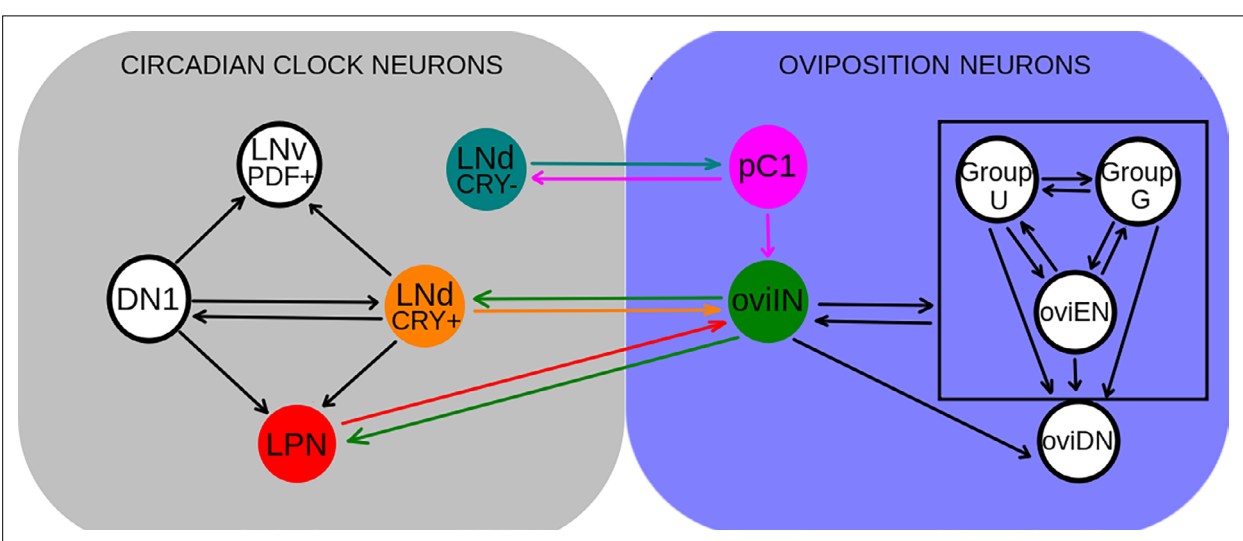

**Figure 6.** Scheme of connections between circadian clock and oviposition-related neuron clusters of the same hemisphere in the hemibrain dataset. Each circle represents a neuron cluster, comprising different numbers of neurons (some clusters comprise only one neuron). Clusters and connections involved in the connectivity between circadian clock and oviposition sets have been colored. In each connection, the arrow points to the post-synaptic cluster. OviIN neurons are bidirectionally connected to every neuron of the clusters inside the square.

the LNds have been shown to be involved in the circadian control of the expression of sex-specific fat body genes (*Erion et al., 2016*; *Fujii et al., 2008*).

The neural circuitry that controls the egg laying process has just begun to be unraveled (*Vijayan et al., 2023*; *Wang et al., 2020*; *Zhou et al., 2014*). It includes the oviDNs, oviINs, oviENs, groups U and G, and pC1 neurons. Using the hemibrain connectome, we found that neither the identified DN1s nor the LNv neurons directly contact any neurons ascribed to the oviposition circuit; the LNd cluster, on the other hand, exhibits synaptic contacts with pC1 and oviIN neurons, providing the anatomical substrate for our experimental results. Furthermore, there is a clear separation of roles inside the LNd cluster, since the oviINs are only contacted by two Cry + LNds (the so-called E1 cells [*Shafer et al., 2022*]) and the pC1s are only contacted by the three Cry- LNds (E3 cells [*Shafer et al., 2022*]). There is also a functional separation within the pC1 cluster, since the only pC1 neurons connected to E3 neurons are the pC1b neurons from both right and left hemispheres (*Figure 5*). As happens with the remaining Cry + LNd neurons, the fifth sLNV does not directly contact any neurons of the oviposition circuit. The resulting clock/oviposition network is summarized in *Figure 6*.

Our data posits the E1 LNds-oviIN connection as the candidate to mediate the timing of egg laying rhythms. The oviIN neurons are two of the most important neuronal hubs included in the current version of the *Drosophila* connectome, receiving strong inputs of hundreds of other neurons. Thus, the oviINs integrate the information from the circadian clock with the information coming from many other sources in order to signal to the oviposition circuit. Under such a scenario, the circadian clock would compete with many other sources to set the threshold for egg deposition (*Vijayan et al., 2023*).

We hypothesize that the E3-pC1b connection is less important for conveying circadian information to the oviposition circuit because the pC1b are located upstream within the oviposition circuit that controls the motor program, being connected only to the oviIN neurons (*Wang et al., 2020*). On the other hand, since pC1 neurons control female receptivity, it is tempting to speculate that this behavior may have a circadian rhythm, that could be generated through the E3-pC1b connection. Interestingly, the female mating rate has been shown to display circadian rhythmicity (*Sakai and Ishida, 2001*), and very recently it has been proposed that pC1b neurons are involved in the control of female copulation rate (*Li et al., 2023*). Conversely, given that the E3-pC1b connection is bidirectional, it could also be instrumental in conveying information about mating status to the circadian clock, therefore, producing the changes in temporal organization that females undergo after mating (*Delbare et al., 2023*; *Isaac et al., 2010*; *Riva et al., 2022*).

**Table 1.** All fly strains used in this study.

| *Drosophila melanogaster* line | Source | Identification number | Common denomination |
|---|---|---|---|
| *Canton-S* | *Bloomington Drosophila Stock Center* | BDSC: 64349 | *CS* |
| *Yellow white* | *Bloomington Drosophila Stock Center* | BDSC: 1495 | *y w* |
| y perS w | Donated by Jeff Hall | Collection of Jeff Hall | per$^s$ |
| y[1] sc[*] v(1) sev[21]; P{y[+t7.7] v[+t1.8]=TRiP.HMS02045}attP2/TM3, Sb[1] | *Bloomington Drosophila Stock Center* | BDSC:40878 | UAS-*per*$^{RNAi}$ |
| w[*]; P{w[+mC]=Clk-GAL4.-856}2 | *Bloomington Drosophila Stock Center* | BDSC:93198 | *Clk856*-Gal4 |
| w[*]; sna[Sco]/CyO; P{w[+mC]=Clk-GAL4.1.5}4.1 M/TM6B, Tb[1] | *Bloomington Drosophila Stock Center* | BDSC:36316 | *Clk4.1*-Gal4 |
| P{w[+mC]=Pdf-GAL4.P2.4}X, y[1] w[*];Dicer/cyo | *Bloomington Drosophila Stock Center* | BDSC:6899 | *PdfDicer*-Gal4 |
| *Mai*179-Gal4;*Pdf*-Gal80 | Donated by José Duhart | Collection of Patrick Emery | *Mai179-Gal4;Pdf-Gal80* |
| MB122B E-cell split-Gal4 | Donated by José Duhart | Collection of Orie Shafer | *MB122B*-splitGal4 |
| w;; kir2.1 (1)/TM3 Sb | Donated by Justin Blau | Collection of Justin Blau | UAS-*kir*2.1 |
| w;UAS-cyc$^{DN}$;+ | Donated by Fernanda Ceriani | Collection of Sebastian Kadener | UAS-cyc$^{DN}$ |

The function of the LPN neurons in the circadian clock is still unclear (*Reinhard et al., 2022*) and, therefore, no clear role in rhythmic oviposition can be ascribed to the LPN-oviIN connection. On the other hand, since there are many more oviIN inputs to the LPN than outputs (135 vs 16 synaptic connections), it seems more likely that this connection conveys information to the circadian clock from those processes in which the oviINs are involved.

The information provided by the *Drosophila* connectome is only the first step towards the understanding of the influence of the circadian clock on oviposition. Future work will be necessary to test the functionality of these connections as well as the role of different neuropeptides (sNPF, ITP, PDF) and neurotransmitters (such as acetylcholine and glutamate) in the control of egg laying behavior.

## Materials and methods

### Fly strains

All fly strains used in this study are detailed in *Table 1*. Flies were reared and maintained on standard cornmeal/agar medium at 25 °C and 60% humidity in a 12 hr:12 hr LD cycle unless stated otherwise. For oviposition and locomotor activity experiments, we used a different medium, named banana medium, prepared with 200 g of banana, 25 g of barley, 36 g of black sugar, 35 g of yeast, 12.5 g of agar, and 2 g of Nipagin per liter of water.

### Semi-automated egg collection device

The egg collection device (*Figure 1—figure supplement 1* and *Figure 1—video 1*) consists of a wooden basis with a mechanical arm fixed to it, and plastic sets of tracks with fly chambers that can be replaced at will. Each 'set of tracks' is a 13.9×19.5 cm 3D-printed piece with seven tracks. Inside each track, there are six equidistant wells measuring 15×20 mm, which are filled with banana medium. Each fly chamber (17×27 mm) has a transparent roof, and no floor, so that the fly inside it is in contact with the food wells, when the chamber is placed on the track so that it can slide along it. Three sets of tracks are placed on the wooden base, and the chambers are moved together from well to well by a mechanical arm, which is moved by a stepper motor controlled by an Arduino UNO. The chambers are displaced from well to well every 4 hr. More frequent sampling gives rise to less consistent rhythmic patterns. Survival in this egg collection device is, in general, high: on average, more than 80% of the flies are alive at the end of an egg collection experiment lasting one week. More information about the egg-laying collection device developed in this study is available upon request.

### Behavioral assays

The egg deposition behavior of females was analyzed individually at 25 °C and 60% humidity. Before starting the assay, six 0-5-day-old virgin females and five males of the desired genotype were anesthetized using $CO_2$ and introduced into a vial with 10 mL of standard food for 72 hr. This allowed for the crossing of 50–80 female flies with their respective males per experiment. Subsequently, the resulting gravid females from the crosses were placed individually in vials containing a plastic spoon with 1.5 mL banana medium along with a drop of yeast for one day. The eggs deposited by each of the females are counted under a binocular stereoscopic magnifier (Lancet Instruments) and the 21 best egg layers are selected to conduct the circadian oviposition experiment. This prior selection of the best egg layers is performed because, of the total females put to mate with males, nearly half do not lay eggs or lay very few.

21 of these females were housed in the chambers of each apparatus, which was then introduced in an incubator at constant temperature (25 °C). The first 1 or 2 days the light regime was LD (12:12) in order for the flies to adapt to their new environment, and then they entered a DD regime for a given number of days. The sets of tracks were retired and replaced by sets with new food every 20, 24, or 28 hr (in order to avoid providing an entrainment signal to the flies). The eggs in each food well of each of the retired tracks were counted under a binocular stereoscopic magnifier (Lancet Instruments), and registered.

For the locomotor activity assays, we used the setup used in previous works (*Riva et al., 2022*). Briefly, the flies are housed in translucid tracks and their movement is registered with video cameras above the tracks, connected to a computer, where the position of each fly is extracted from the video (with ad hoc programmed software) and registered. Locomotor activity of mated females was

monitored for 3 days in LD conditions and then transferred to DD conditions for 7 days. Every 4 or 5 days, the flies were transferred to tracks with fresh food, because after that time, the appearance of larvae hindered the video tracking of the females.

## Data analysis

After each oviposition experiment lasting N days, the data consisted of a time series *E* of 6 N points for each fly. A new time series was generated by averaging the individual series. Since the number of eggs laid by a mated female tend to show a downward trend, we proceeded as follows, in order to detrend the data (see the Appendix for further details). First, a moving average of the data is performed, with a 6 point window, and a new time series *T* is obtained. In principle, *T* is a good approximation to the trend of the data. Then, a new, detrended, time series *D* is generated by pointwise dividing the two series (i.e. *D(i)=E(i)/T(i)*, where *i* indexes the points of each series).

In order to assess the rhythmicity of a genotype, we averaged the detrended time series and performed a Lomb-Scargle periodogram (*Ruf, 1999*; *VanderPlas, 2018*). For this, we used package Lomb (version 2.1.0), from the R Statistical Software (v4.1.2; R Core Team 2021). The significance lines were calculated by repeatedly randomizing the time series (using function *randslp* in Lomb R package). A genotype was considered rhythmic if the periodogram had one peak between 16 and 32 hs, and it was above the *p*=0.05 significance line.

The graphs showing individual periods were made by obtaining the individual periods using the Lomb-Scargle periodogram described above for each particular individual. For the individual periods shown in the figures, we added the weakly rhythmic category, corresponding to flies whose periodograms displayed only one peak between 18 and 32 hs, and a power larger than 0.2.

More details about the method used can be found in Appendix 1.

For locomotor activity experiments, these files obtained from the video tracking were processed with an analysis software we developed (in Bash) which provides statistics for activity (position, distance traveled, etc.) (*Riva et al., 2022*).

## Analysis of connectome data

The data used to determine the connectivity between the circadian clock and the oviposition circuits come from the Hemibrain dataset (version 1.2.1) made publicly available by Janelia Research Campus (*Scheffer et al., 2020*). To access the data, we used the NeuPrintExplorer Web tool (https://neuprint.janelia.org/). The result of our queries is summarized in *Supplementary file 1B and C*. The neuron IDs are as given in the hemibrain dataset, just as the instance names of the oviposition-related neurons. For circadian clock neurons, we used the names given by *Shafer et al., 2022*. We only replaced LPN-4 by LPN-L to stress the fact that this neuron is placed in the left hemisphere (all the others are in the right hemisphere). The strength of each connection was quantified using the criteria of *Scheffer et al., 2020*. Connections with more than 9 synapses were considered strong, connections having between 3 and 9 synapses were considered as having intermediate strength, and connections with 0, 1, or 2 synapses were considered as weak. Because weak connections are prone to error (*Scheffer et al., 2020*), we excluded them from the analyses. The figures that show connectivity between sets of neurons were drawn using Cytoscape version 3.10.1 (*Shannon et al., 2003*).

## Statistical analysis

The following statistical analyses were used for the comparison between genotypes of individual periods (scatter plots). First, we tested data for normality with D'Agostino & Pearson test. Once the normality of the data was confirmed, we used an unpaired t-test for test differences between means for experiments where two groups were compared and a one-way ANOVA for experiments where three groups were compared. When data was not normally distributed, we applied a Mann-Whitney test for test differences between means. When significant differences were found, we conducted a post-hoc Tukey test with a correction for multiple comparisons. In all the analyses, we used two-tailed *p*-values. To perform all the tests, we used GraphPad Prism (Boston, Massachusetts USA, https://www.graphpad.com) and R Statistical Software (v4.1.2; R Core Team 2021). *p*<0.05 was considered statistically significant. In dot plots, horizontal lines represent the mean value; error bars depict the standard error of the mean. For the comparison between genotypes of the proportion of rhythmic individuals over the total (proportion plots), we used the 'N-1' Chi-squared test as recommended

by *Campbell, 2007* and *Richardson, 2011*. To perform these tests, we use MedCalc Software Ltd. Comparison of proportions calculator https://www.medcalc.org/calc/comparison_of_proportions.php (Version 23.2.7; accessed June 19, 2025). No statistical methods were used to determine sample size. Sample sizes are similar to those generally used in this field of research. Samples were not randomized and analyzers were not blind to the experimental conditions. In all the figures, we show results of two or three independent experiments.

## Acknowledgements

This work was supported by the Agencia Nacional de Promoción Científica y Tecnológica (Grants PICT-2016-1042 to SRG; PICT PICT2019-1015 to MFC and PICT-2021-00213 to DLF), the Universidad Nacional del Comahue, CRUB (Grant 04/B239 to DLF). DLF, SRG, and MFC are members of the Argentine Research Council for Science and Technology (CONICET). SR holds a graduate fellowship from CONICET. Stocks obtained from the Bloomington *Drosophila* Stock Center (NIH P40OD018537), were used in this study. We thank Dr. Lucas Mongiat for helpful discussions and critical reading of the manuscript.

## Additional information

### Funding

| Funder | Grant reference number | Author |
| --- | --- | --- |
| Agencia Nacional de Promoción de la Investigación, el Desarrollo Tecnológico y la Innovación | PICT 2016-1042 | Sebastián Risau-Gusman |
| Agencia Nacional de Promoción de la Investigación, el Desarrollo Tecnológico y la Innovación | PICT2019-1015 | Maria Fernanda Ceriani |
| Agencia Nacional de Promoción de la Investigación, el Desarrollo Tecnológico y la Innovación | PICT-2021-00213 | Diana Lorena Franco |
| Universidad Nacional del Comahue, CRUB | Grant 04/B239 | Diana Lorena Franco |

The funders had no role in study design, data collection and interpretation, or the decision to submit the work for publication.

### Author contributions

Sabrina Riva, Formal analysis, Investigation, Methodology, Writing – original draft, Writing – review and editing; Maria Fernanda Ceriani, Supervision, Funding acquisition, Writing – review and editing; Sebastián Risau-Gusman, Conceptualization, Software, Formal analysis, Supervision, Investigation, Methodology, Writing – original draft, Writing – review and editing; Diana Lorena Franco, Conceptualization, Formal analysis, Supervision, Funding acquisition, Investigation, Methodology, Writing – original draft, Project administration, Writing – review and editing

### Author ORCIDs

Sabrina Riva ⬤ https://orcid.org/0000-0002-3671-828X
Maria Fernanda Ceriani ⬤ https://orcid.org/0000-0001-8945-3070
Sebastián Risau-Gusman ⬤ https://orcid.org/0000-0003-2475-4547
Diana Lorena Franco ⬤ https://orcid.org/0000-0002-2752-5603

Joint Public Review: https://doi.org/10.7554/eLife.103359.4.sa1

Author response https://doi.org/10.7554/eLife.103359.4.sa2

## Additional files

### Supplementary files
MDAR checklist

Supplementary file 1. Tables of connections given in the connectome from clock neurons to oviposition neurons (Table S2), and from oviposition neurons to clock neurons (Table S3).

### Data availability
All data is now available at https://github.com/srisaug/flywork (copy archived at *Risau-Gusman, 2026*).

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

# Appendix 1

## Method used for the assessment of rhythmicity

The egg records of single, isolated, female fruit flies display several distinct features that complicate the assessment of rhythmicity. The recorded variable is discrete and typically takes only a few values, which depend on how often it is sampled. The more often it is sampled, the less number of values the variable takes. Thus, it is only possible to sample it a few times (usually less than ten) per cycle. As shown below, for noisy signals, this feature tends to make the assessment of rhythmicity much more complicated, compared with signals with the same levels of noise, but that are sampled more often (as happens with activity records).

The particularities of egg-laying in *Drosophila* add further complications. First, the number of eggs laid has a clearly descending trend (probably due to the depletion of the sperm stored in the female spermatheque). Second, this behavior seems to be particularly 'noisy,' in the sense that the decision of laying an egg seems to depend not only on the time of the day, but also on many other stimuli. As a consequence, any method that tries to assess the presence of a rhythm must be able to subtract, at least partially, the effects of trend and noise. In the following, we describe the method we have used to achieve this aim.

## Noise effects and subtraction

We use Lomb-Scargle (LS) periodograms (*Ruf, 1999*; *Zielinski et al., 2014*) to determine if a signal is rhythmic, and to estimate its period. It is important to note that, as happens with Discrete Fourier Transforms, periodograms only give information about a number of predetermined periods (usually called the 'frequency grid'), and the number of available periods (and therefore the resolution with which the true period of the signal can be ascertained) is proportional to the time length of the signal. The significance lines are obtained by repeatedly shuffling the signal and recording the highest peaks of the resulting periodograms. Thus, a *p=0.05* significance line at a given power means that in *95%* of the shufflings the highest peak was below the line.

In order to understand the effect on the assessment of rhythmicity of having few observations per cycle, we have generated noisy harmonic signals and, for the same signal, we have taken different numbers of 'observations' (i.e. we have sampled the signals with different rates), and we have compared the results of period determination in each case. For a signal with period T=24 with Gaussian noise added, panels A and C of *Figure 1* show an example of observations made every 30 min (A), which corresponds to a sampling rate usual in activity records (*Pfeiffenberger et al., 2010*) or every 4 hr (B), more typical of oviposition records. Even though in the corresponding LS periodograms (panels B and D, respectively) the highest peak is at *T=24* h, the power of this peak ($p_P$) is smaller than the power $p_\sigma$ corresponding to a significance of $\sigma=0.05$ (i.e. the peak is not significant) when the number of observations is small, whereas it is much bigger than $p_\sigma$ when the number of observations is larger. Repeating this experiment for different numbers of observations (panel E), confirms that, on average, sampling a signal less often (i.e. taking less observations per cycle) strongly decreases the significance of the peaks obtained.

Since oviposition records have few observations per cycle, we have resorted to the method of taking population averages (*Semmlow, 2018*), in order to get rid of some of the noise. This implies assuming that the underlying periodic signal is basically the same for all individuals, in terms of period and phase. This seems to be a reasonable assumption since activity records consistently show that, after lights off, the periods of activity and sleep tend to be synchronized, at least during the first days.

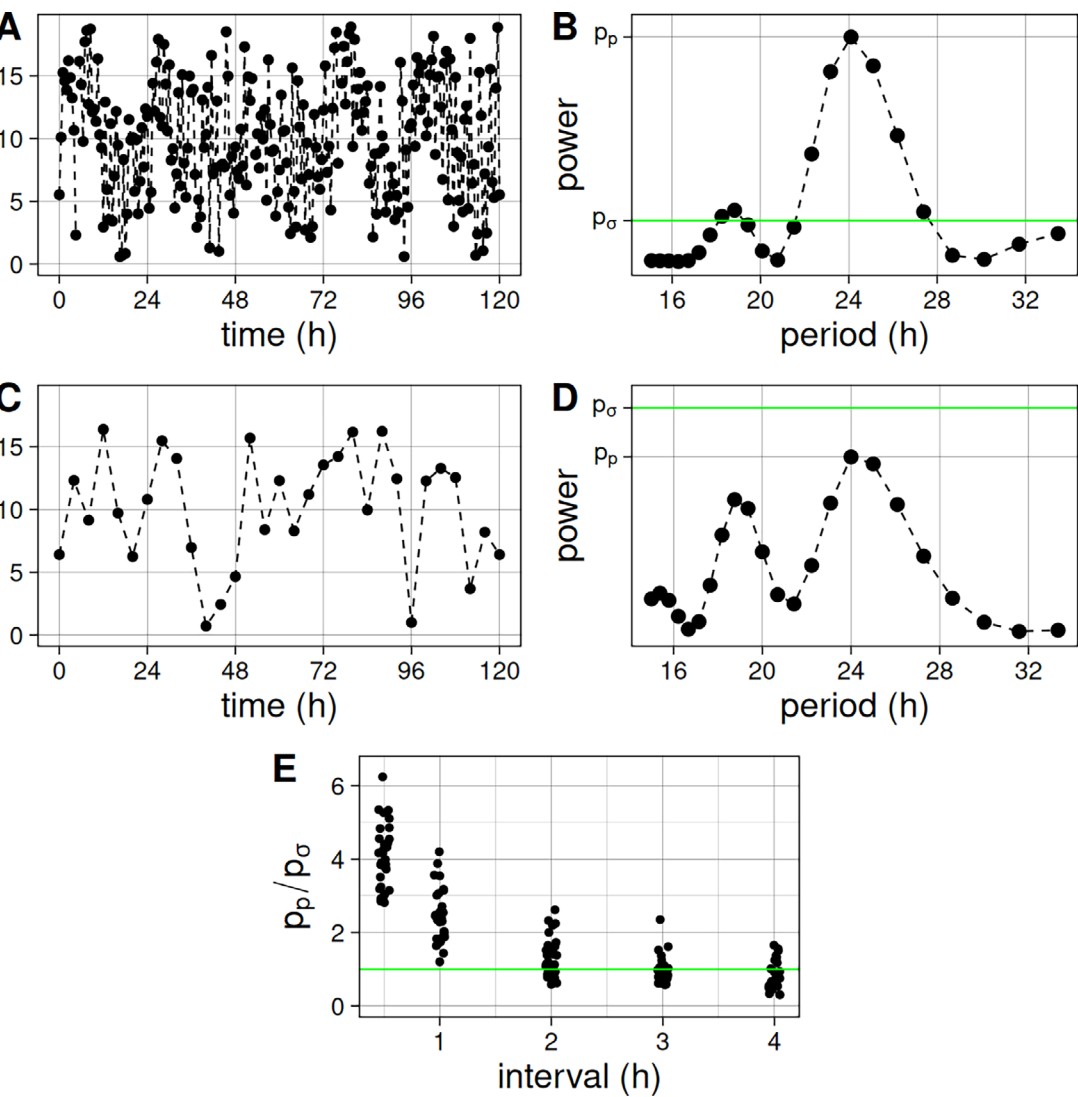

**Appendix 1—figure 1.** Effect of noise in the assessment of rhythmicity of periodic signals. (**A,C**) Periodic signal with Gaussian noise (average=0, sd=2), sampled at intervals of 30 min (**A**) or 4 hr (**C**). (**B, D**) Periodograms of the signals shown in panels A and C, respectively. (**E**) Comparison of periodogram peak heights (relative to the significance level) for 150 signals sampled at different intervals. $p_P$ is the power of the periodogram peak, whereas $p_\sigma$ denotes the power corresponding to a significance level s=0.05. The green line, corresponding to $p_s=p_\sigma$, separates experiments where the peaks are significant (above) from those where peaks are not significant (below).

## Trend subtraction

When a continuous signal is considered as the product $h(t)=f(t)g(t)$ of a periodic signal $f(t)$ with a known period $T$, and a trend $g(t)$ that has little variation over a cycle, the classical method to uncouple them involves using moving averages (see, e.g. *Hyndman and Athanasopoulos, 2021*):

$$g^*(t) = \frac{1}{T} \int_{t-T/2}^{t+T/2} h(t')\, dt'$$
$$f^*(t) = h(t)/g^*(t) \tag{1}$$

where $g^*(t)$ and $f^*(t)$ are the estimates of the trend and the periodic signal. When the assumptions mentioned are met, the estimates coincide with the original functions.

In our case, however, many things are different. The most important are that we do not know what is the period of the 'periodic' signal (in fact, we want an estimate of $T$ if the signal is periodic at all), and that the signal is not continuous. The first feature implies that we ignore what is the 'correct'

value for the size for the moving average. The second feature forces us to convert the moving average integral into a sum with few terms. As a consequence, for a time series of the form *f(0)*, *f(4), f(8),...*, representing observations on the number of eggs taken every 4 hr, we use the following equations for estimating the trend and the underlying periodic signal:

$$g^*\left(t_i\right) = \frac{1}{2n+1} \sum_{j=i-n}^{i+n} h\left(t_j\right)$$
$$f^*\left(t_i\right) = h\left(t_i\right)/g^*\left(t_i\right)$$

(2)

where $t_i$ is the time at which the *i*-th observation was made. We have used n=3, corresponding to a moving average of roughly 28 hr (using n=2, i.e. 20 hr, gives very similar results).

In order to test our method, we apply it here to 'synthetic' egg records, generated with the following model. We assume that there is an underlying signal *f(t)* of period *T* (time is measured in hours), given by

$$f\left(t\right) = \left(1 + sin\left(0.1 + 2\pi t/T\right)\right)$$

(3)

with *A=20* (maximum of eggs at each time point), and that this is corrupted by some additive noise *n(t)*. For simplicity, *n(t)* is simply a random number uniformly chosen between 0 and *A*. The resulting signal is modulated by a decay given by *g(t)=exp(−t/50)*. The total signal is then

$$h\left(t\right) = \text{int}\left[g\left(t\right)\left(alpha f\left(t\right) + \left(1 - alpha\right) n\left(t\right)\right)\right], t = 0, 4, 8, ..., 120.$$

(4)

*Appendix 1—figure 2* shows a graphical example of the partial detrending of a synthetic signal. Panels A and B show, respectively, the synthetic decay and noisy rhythmic signal, with period *T=23.8*, which give the signal shown in panel D, when combined. Notice that when a periodogram of the signal in B is generated, the peak is close but not exactly in 23.8 hr, simply because this frequency is not available in the frequency grid. The periodogram of the full signal (panel E) shows that the decay complicates the determination of rhythmicity, since the peak is now under the significance line. Panels F and G show our estimates for the decay and the noisy rhythmic signal obtained with our method. The periodogram of the signal in G (panel H) confirms that it is rhythmic and with the same period as the original (panel B).

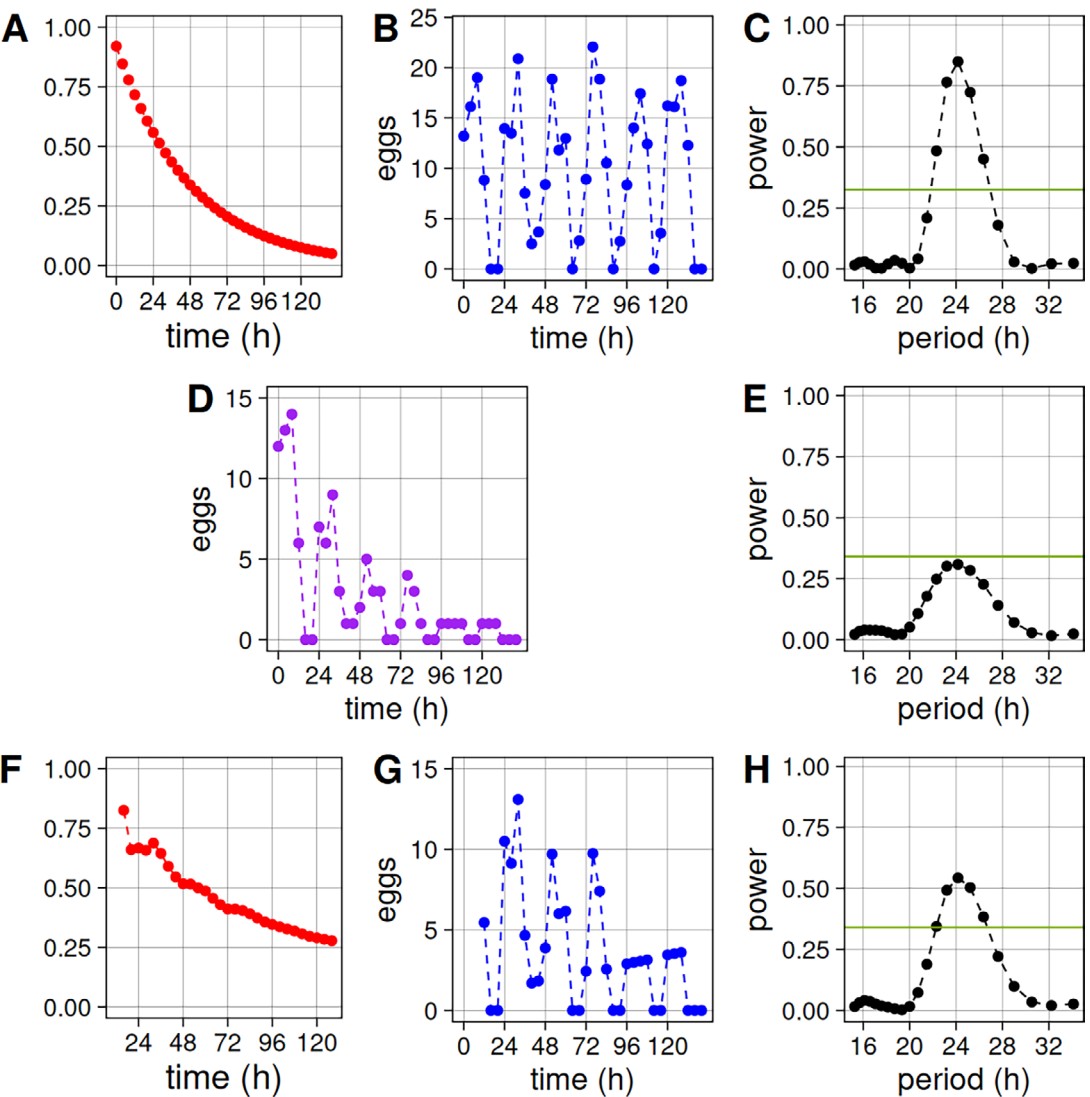

**Appendix 1—figure 2.** Generation and decomposition of a decaying noisy rhythmic synthetic signal.
(**A**) Modulating decay function. (**B**) Noisy rhythmic signal. (**D**) Signal given by the product of signals A and B.
(**F**) Estimate of the decay function. (**G**) Estimate of the rhythmic signal. (**C, E, H**) Periodogram analysis of the
signals in B, D, G, respectively. Green horizontal lines mark the power corresponding to a significance level of *0.05*.
Dashed lines are only guides to the eye.

Even though the size of the moving average window is fixed, the method is effective for assessing
rhythmicity for decaying signals of very different periods, as shown in *Appendix 1—figure 3*.
Furthermore, when the signal is not rhythmic, the method gives a non-rhythmic estimate (panel D
of *Appendix 1—figure 3*). In order to check the accuracy of method estimation we generated 2000
synthetic signals with *T* between 19 and 28 hr (*Appendix 1—figure 4*). First, it must be noticed
that, even before adding decay, the estimation of period is not perfect, because of noise and the
small number of cycles analyzed (panel A). When decay is added, the peaks in the periodogram are
usually under the line of significance. Furthermore, if we use the position of these peaks as estimates,
they tend to be clearly biased to larger values (panel B). When our method is applied, the estimates
obtained are reasonably good, and there is almost no bias (panel C).

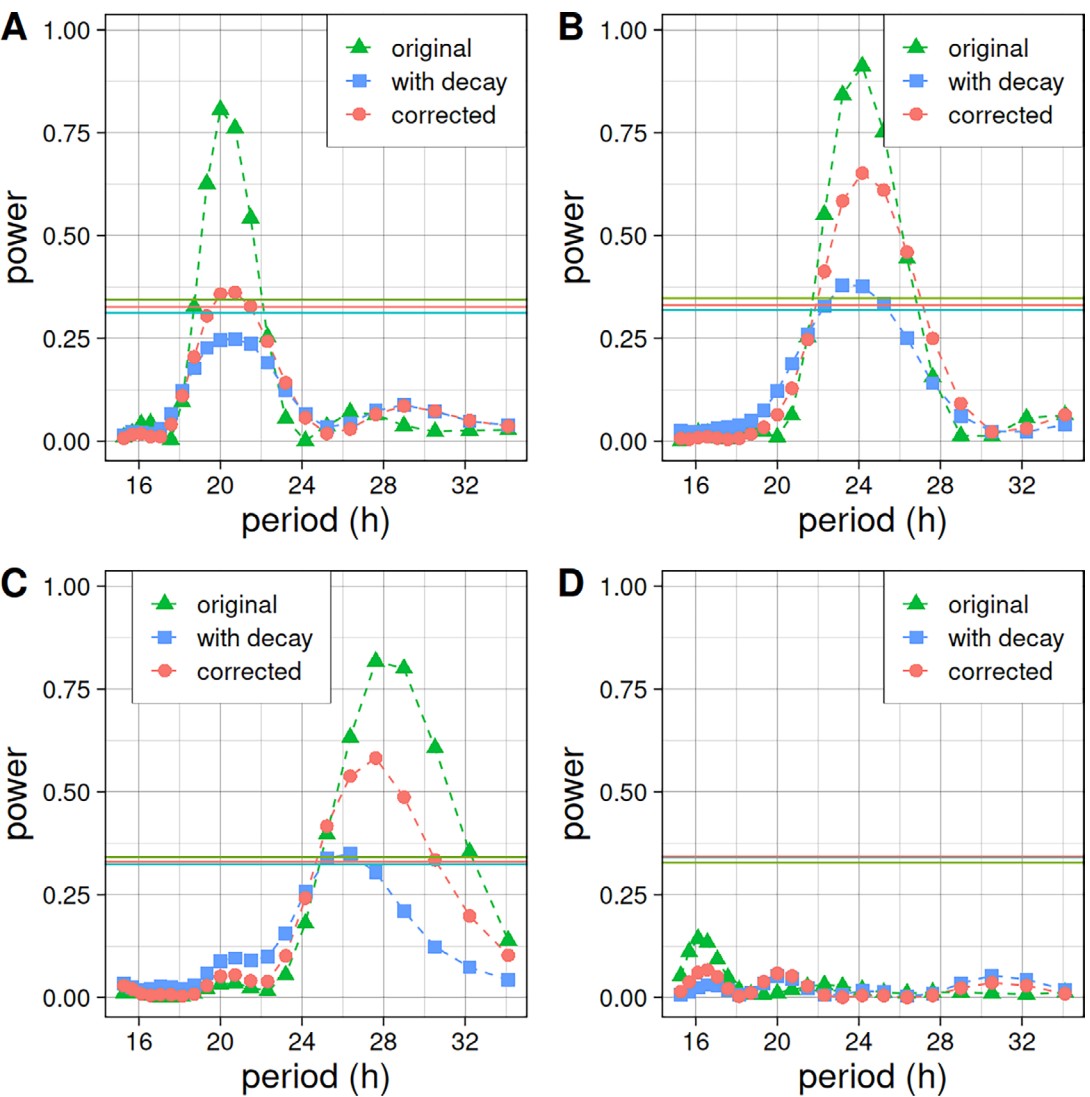

**Appendix 1—figure 3.** Period determination for different synthetic signals. Periodograms of rhythmic noisy signals with period *T* (green triangles), of the same signal with decay applied (blue squares), and of the estimation of the signal (red circles). (**A**) *T=20*. (**B**) *T=24*. (**C**) *T=28*. (**D**): α=0 (i.e. purely arrhythmic signal). Each horizontal line corresponds to the significance level of *0.05* for the periodogram with matching color. Dashed lines are only guides to the eye.

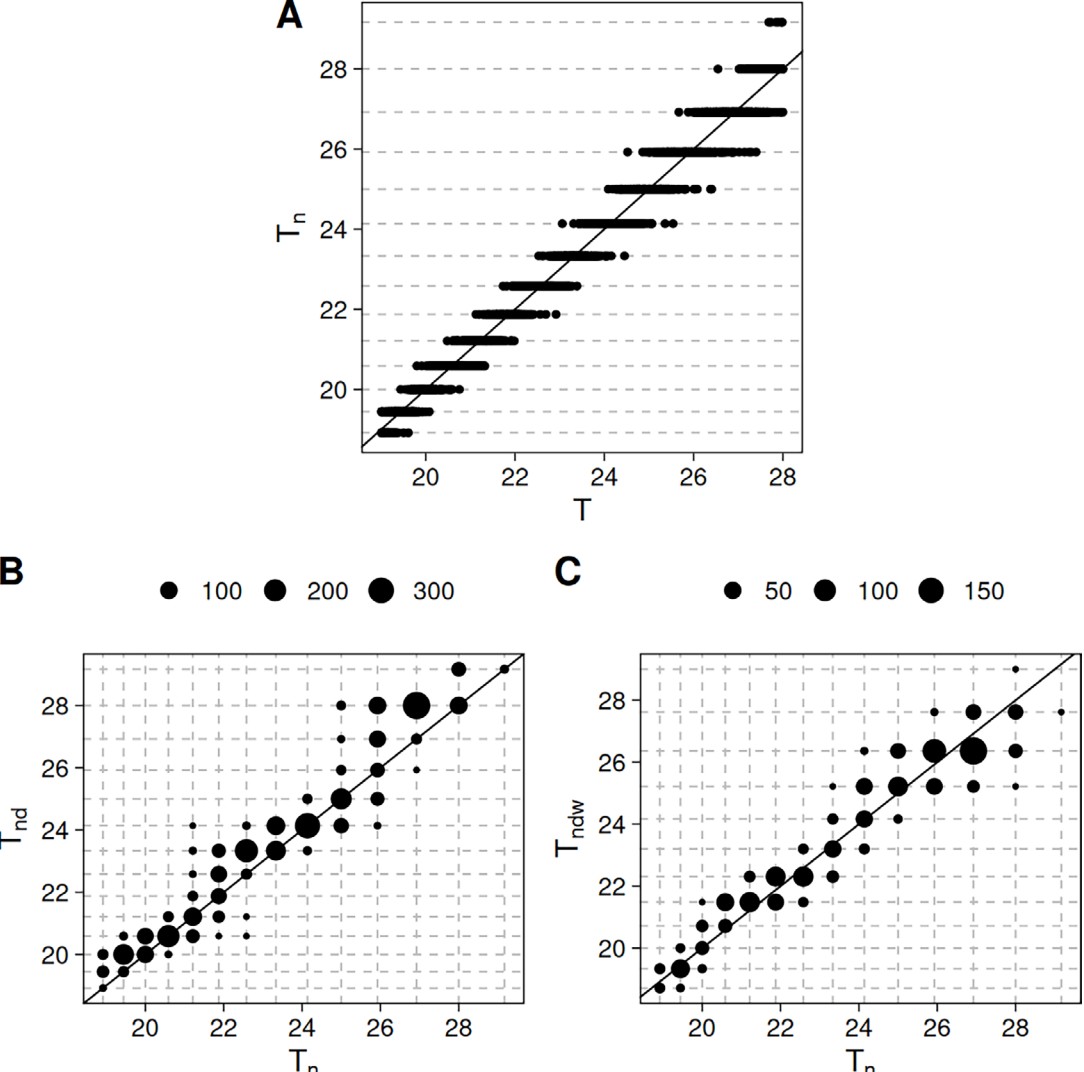

**Appendix 1—figure 4.** Accuracy of period determination for different synthetic signals. (**A**) Period $T_n$ estimated for 2000 noisy rhythmic signals generated with $T$ between 19 and 28 hr. (**B**) Period $T_{nd}$ estimated for the same signals as in A when decay is applied, as a function of the period estimate $T_n$ of the original signal. (**C**) Period $T_{ndw}$ estimated when our method is applied to the signals in B, as a function of the period estimate $T_n$ of the original signal. Horizontal and vertical dashed lines correspond to the values present in the corresponding period grid. In B and C, the size of the symbols is proportional to the number of signals with the periods indicated by the crossing grid lines.

In order to evaluate how good is our method is to detect rhythmicity in a signal, independently of the period, we analyzed the p-values obtained for 200 rhythmic and arrhythmic signals. First, *Appendix 1—figure 5* shows that, after decay is added, the p-values of almost all peaks in the periodograms are above $p=0.05$, which implies that they would be considered as arrhythmic. When our method is used to eliminate, at least partially, the decaying component, the remaining signals have p-values smaller than 0.05, and would, therefore, be classified as rhythmic, in 55% of the cases. In other words, there is a 45% of false negatives. This is to be expected since in some cases the combination of noise and decay render almost impossible the recovery of the rhythmic signal. On the other hand, panel B of *Appendix 1—figure 5* shows that, in this case, the method has no false positives since none of the arrhythmic signal has been classified as rhythmic after detrending is performed.

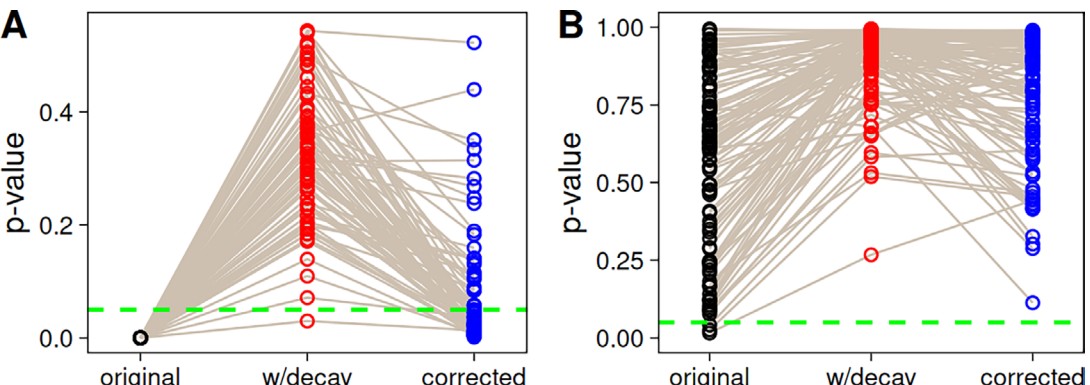

**Appendix 1—figure 5.** Presence of false negatives and absence of false positives. P-values obtained for 100 rhythmic (**A**) and 100 arrhythmic (**B**) synthetic signals. Left: Noisy rhythmic signals. Center: Signals obtained by applying decay to the previous signals. Right: Signals obtained by applying our method to the previous ones. The green dashed line indicates the level of significance used (0.05).

## Example of rhythmicity assessment for a synthetic experiment

As mentioned above, when noise is strong, detrending may not be enough, and one must resort to population averaging. To see how well our method performs in this setting, we mimimicked a rhythm assessing experiment on a group of flies by generating 28 rhythmic individual records of 6 days lengths, using the model described above. The periods $T$ for each record were drawn from a Gaussian with average 23.8 hr and 0.1 hr of statistical deviation. The resulting records are shown in *Appendix 1—figure 6*. After the application of our detrending, one can notice that, out of 28 flies, only 5 would be classified as rhythmic (see *Appendix 1—figure 7* for the detrended records and *Appendix 1—figure 8* for the corresponding periodograms). However, when a population average is performed, the resulting periodogram reveals the rhythmicity of the population (*Appendix 1—figure 9*).

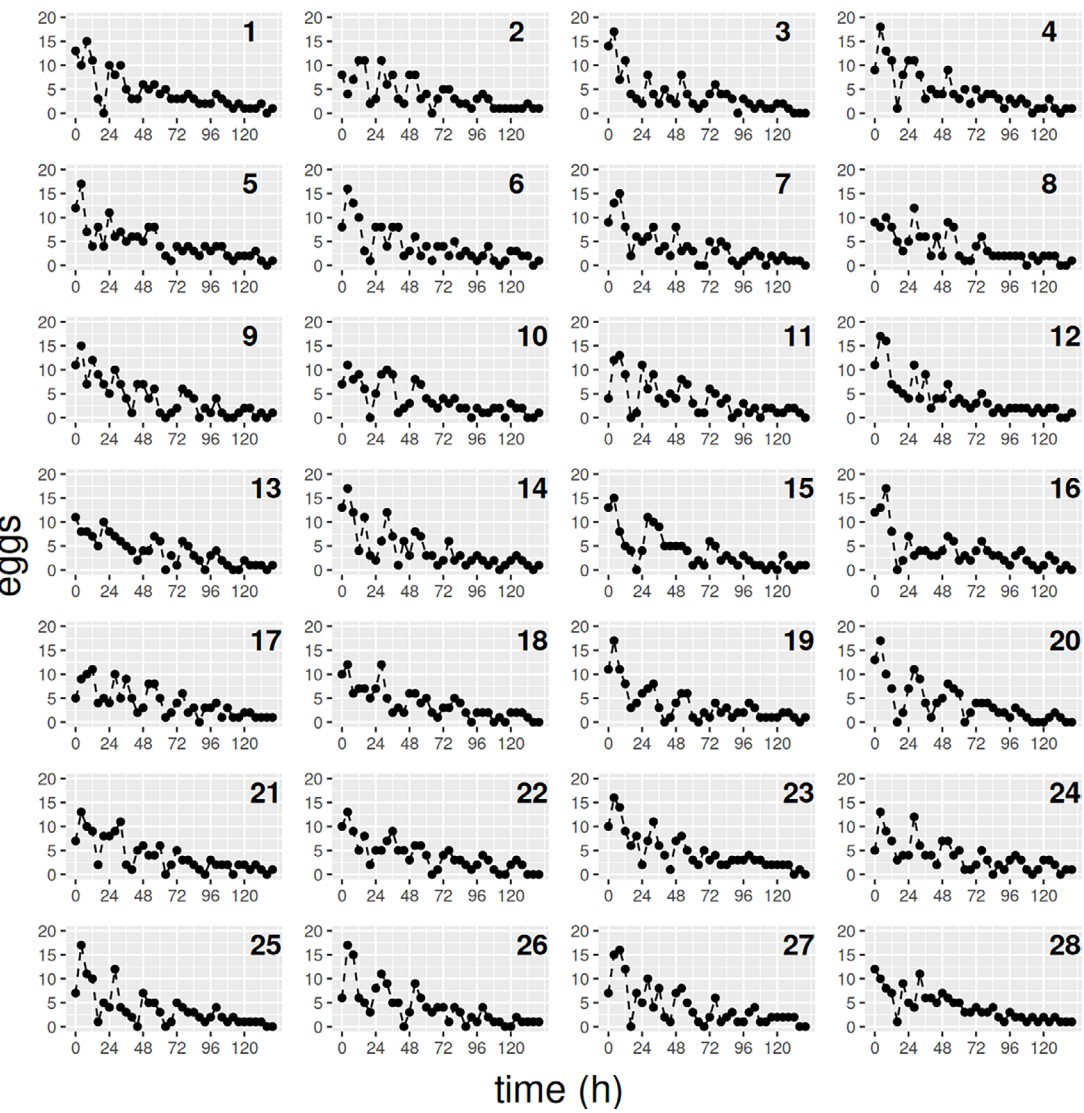

**Appendix 1—figure 6.** Individual egg records for the first synthetic experiment with 28 flies.

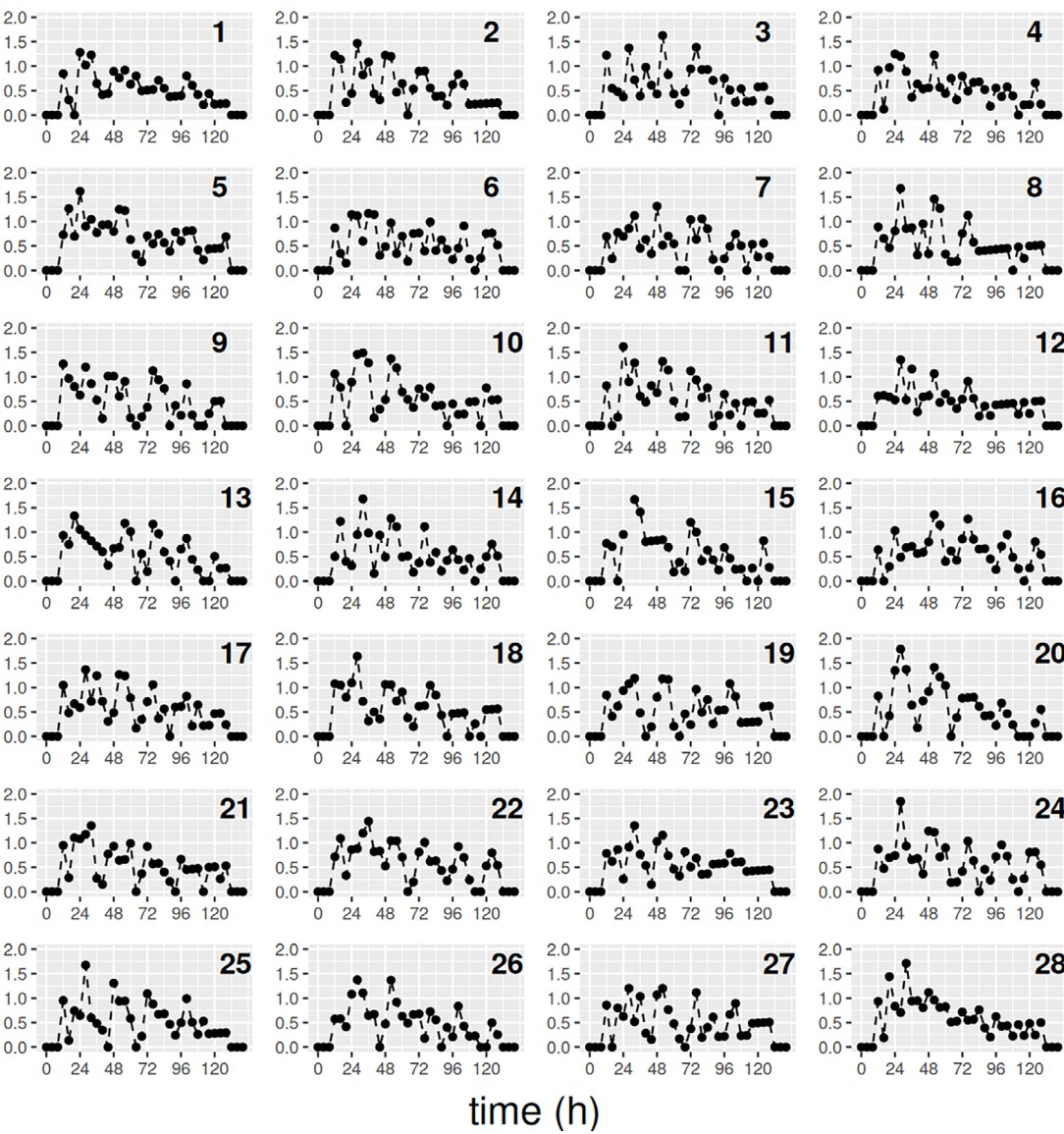

**Appendix 1—figure 7.** Records obtained after applying detrending for the first synthetic experiment described in the text.

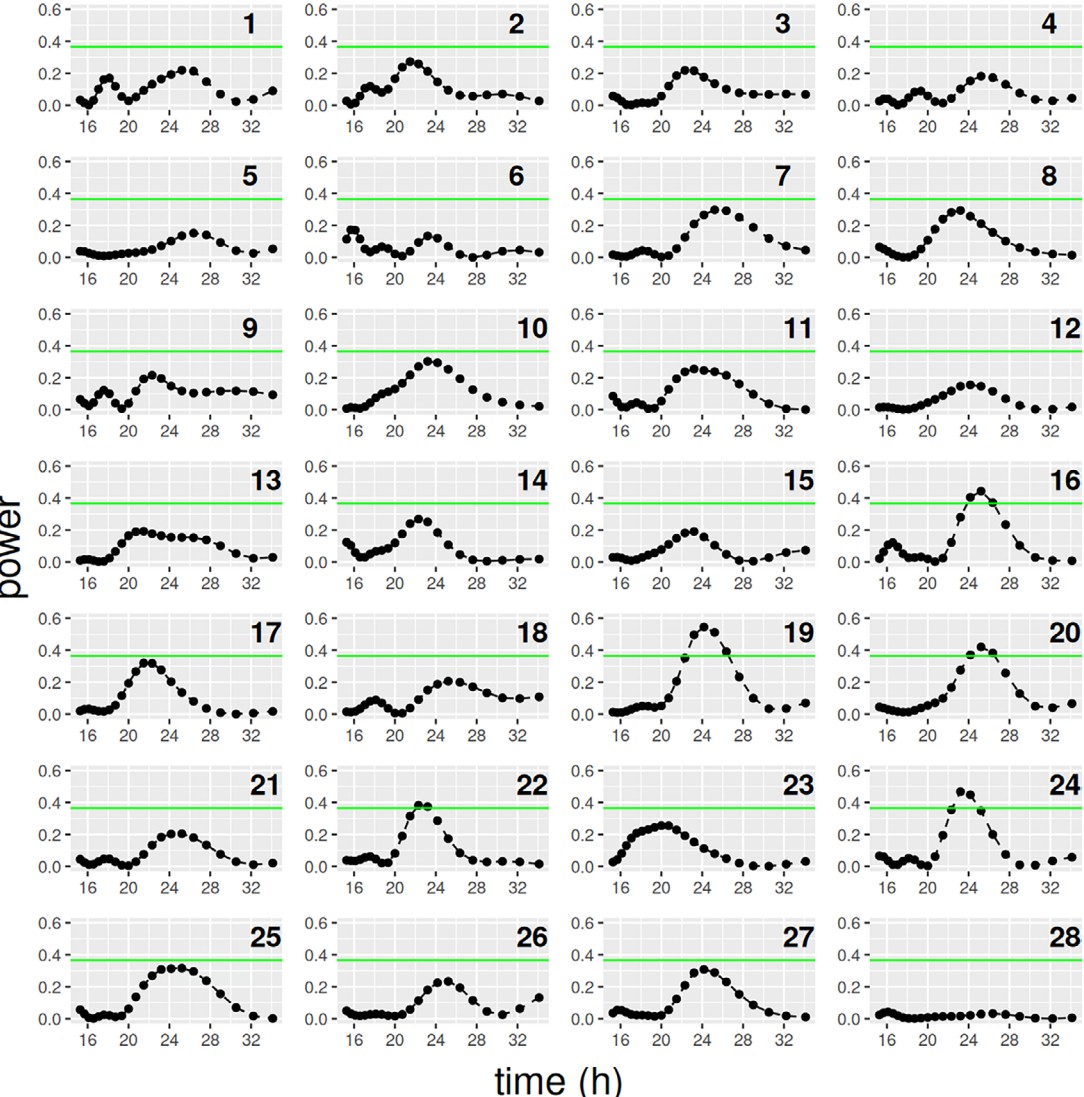

**Appendix 1—figure 8.** Periodograms for the records obtained after applying detrending for the first synthetic experiment described in the text.

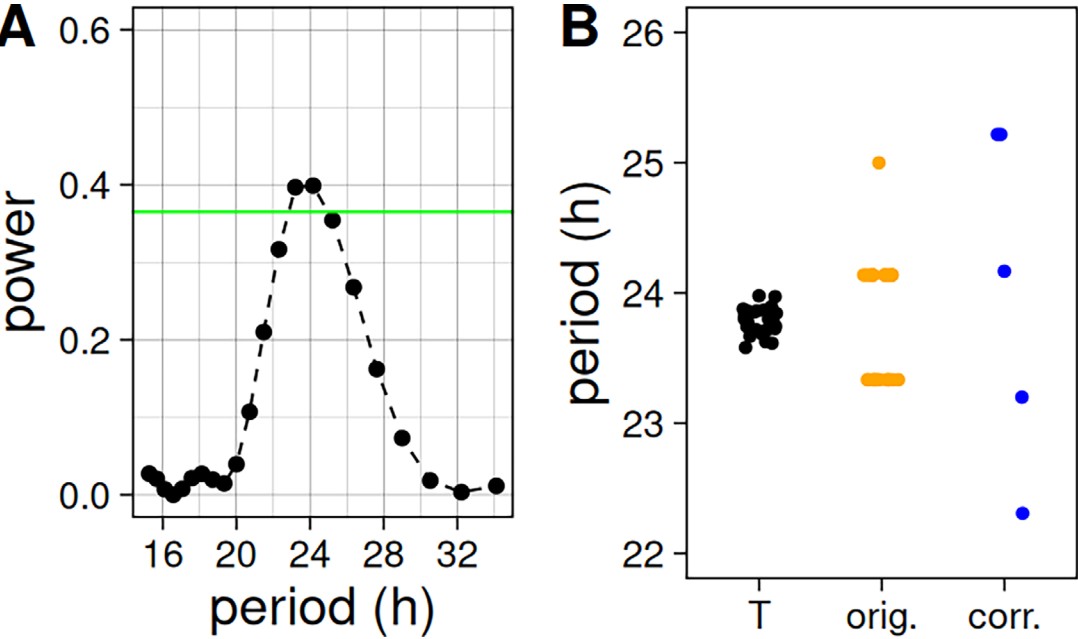

**Appendix 1—figure 9.** Results for the first synthetic experiment described in the text. (**A**) Population periodogram. (**B**) Individual periods (Left: *T* of the underlying rhythm, center: period estimate of the signal without decay, right: estimated periods of the records with a significant peak in the periodogram).

As a control, we performed a second synthetic experiment, but now with completely random individual 'flies,' obtained by using *α=1* in *Equation 4*. The individual and detrended records are displayed in *Appendix 1—figures 10–12*. The population periodogram (*Appendix 1—figure 13A*) shows clearly that the population is arrhythmic. When the population classified as rhythmic, but there are only a few individuals that display a clear rhythmicity, it could be objected that the population rhythm is in fact driven by these very rhythmic individuals. In order to test this possibility, we eliminated the five rhythmic records of the first experiment and analyzed the group of the remaining flies. The resulting periodogram is displayed in *Appendix 1—figure 13B*. Even though the peak is now lower, the rhythmicity is weaker but still present.

Conversely, when 5 random flies of our second experiment (where all flies were truly arrhythmic) are replaced by the five rhythmic flies of the first experiment, the population periodogram obtained does not reveal a peak in the population (*Appendix 1—figure 13C*).

## Example of rhythmicity assessment for a real experiment

In order to check all this in a real experiment, we analyzed the records from two experiments for CantonS and per01 flies. The experiments were performed in DD conditions, using the protocol explained in the main text. The individual records and their corresponding periodograms are shown in *Appendix 1—figures 14–19*. From *Appendix 1—figure 14*, we can check that only seven flies are individually rhythmic. The population average, however, is strongly rhythmic (see *Appendix 1—figure 20A*). We have also assessed the rhythmicity when these individually rhythmic flies are eliminated from the sample. The population average is still rhythmic, albeit with a much reduced power, as was to be expected. It is interesting to compare this with the results obtained for per01 flies, which have been shown to lack a functional circadian clock. In this case, the average is very arrhythmic, even though one fly appears to be individually rhythmic.

These results confirm that our method can be used to detect rhythms that are usually hidden behind the decay of the signal, and are also strongly contaminated by behavioral noise.

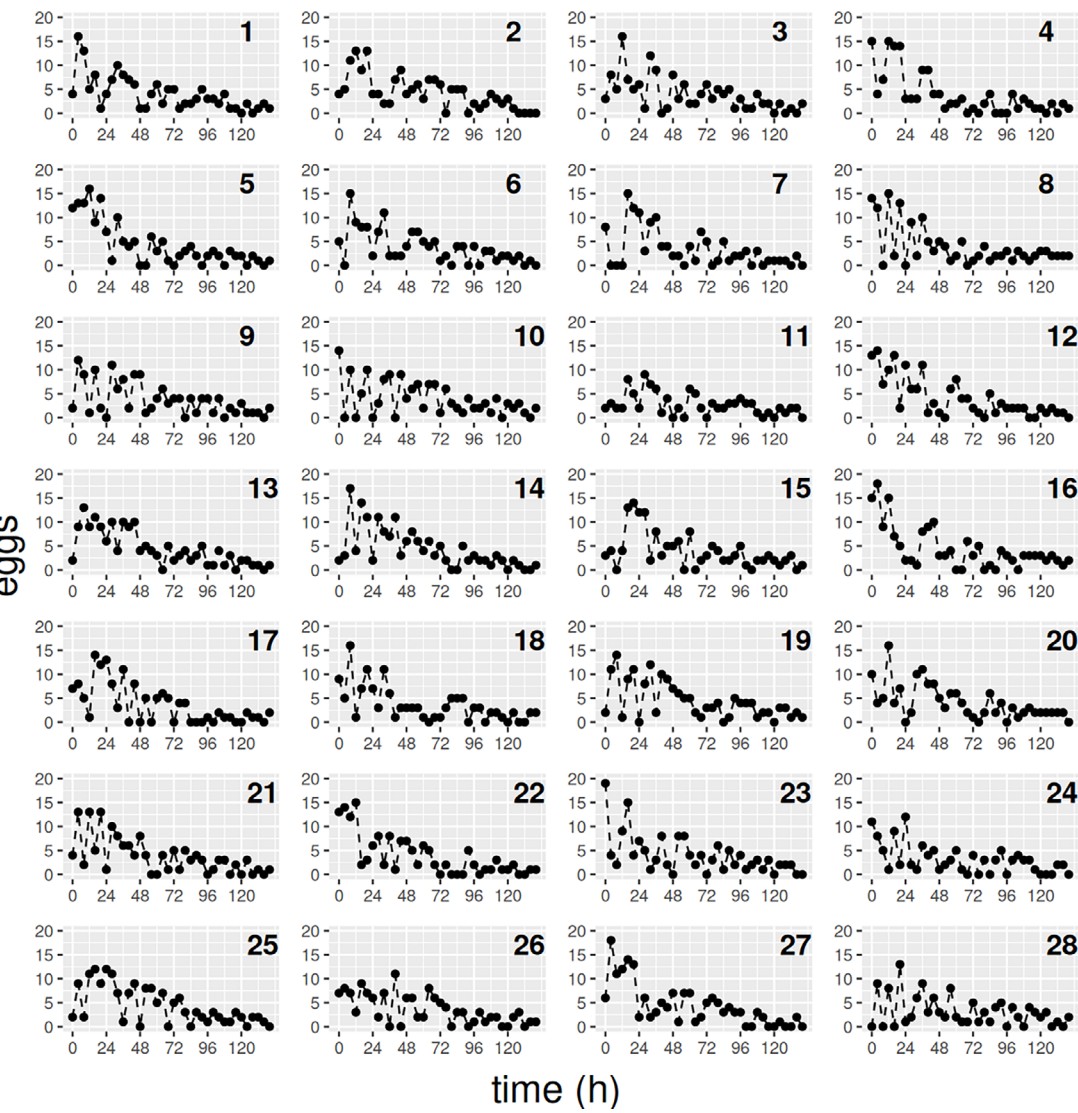

**Appendix 1—figure 10.** Individual egg records of the second synthetic experiment with 28 flies.

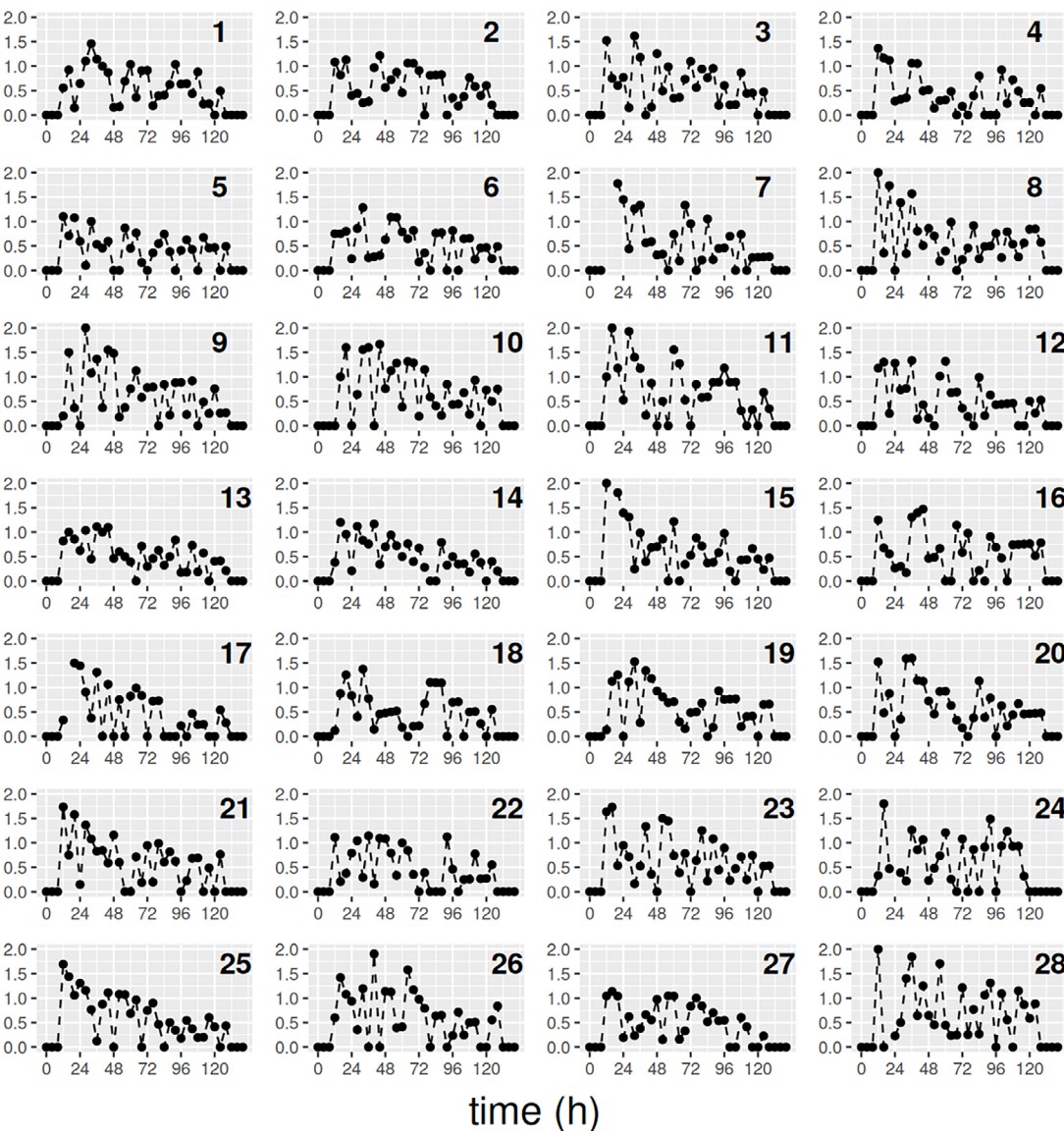

**Appendix 1—figure 11.** Records obtained after applying detrending for the second synthetic experiment described in the text.

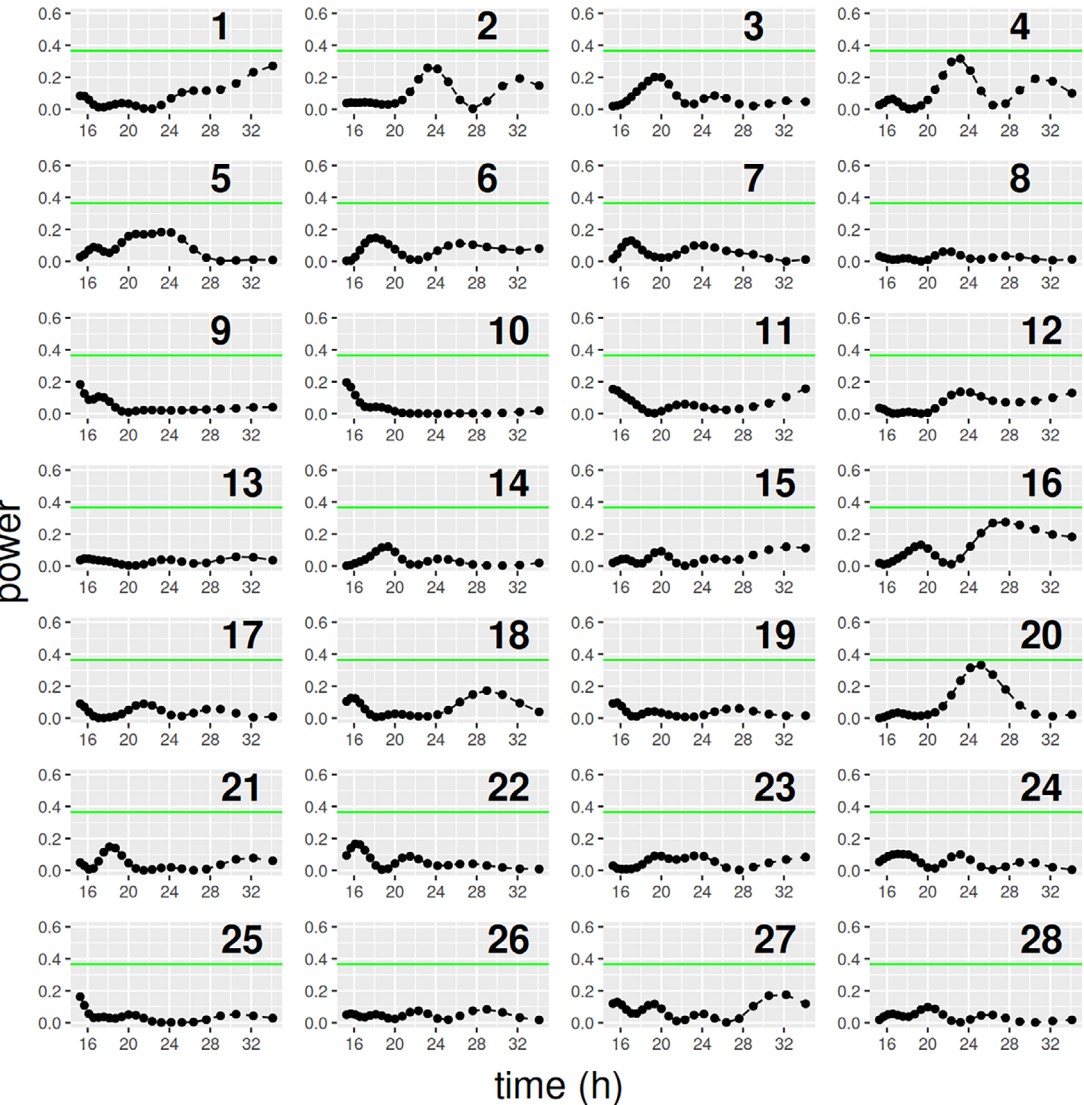

**Appendix 1—figure 12.** Periodograms for the records obtained after applying detrending for the second synthetic experiment described in the text.

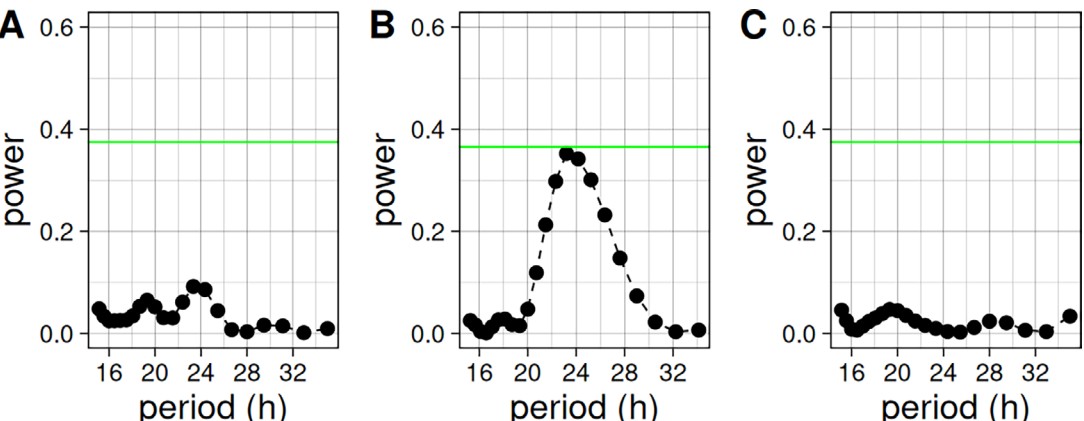

**Appendix 1—figure 13.** A few rhythmic flies do not make a population rhythmic. (**A**) Population periodogram for the second experiment mentioned in the text. (**B**) Population periodogram for the first experiment without the five most rhythmic flies. (**C**) Population periodogram for the second experiment with five flies replaced by the five rhythmic flies of the first experiment.

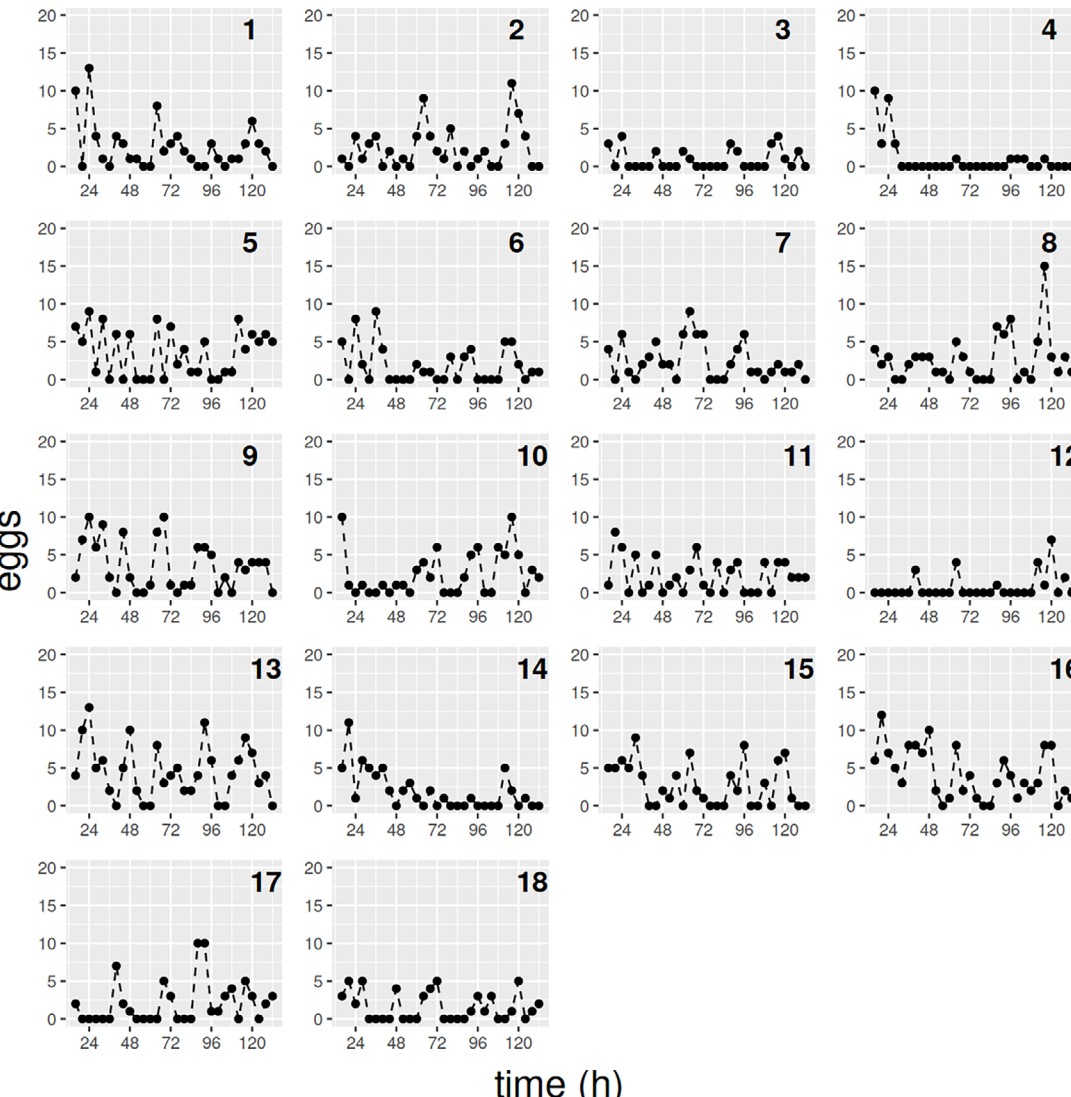

**Appendix 1—figure 14.** Individual egg records of an experiment in DD with 18 CantonS female flies.

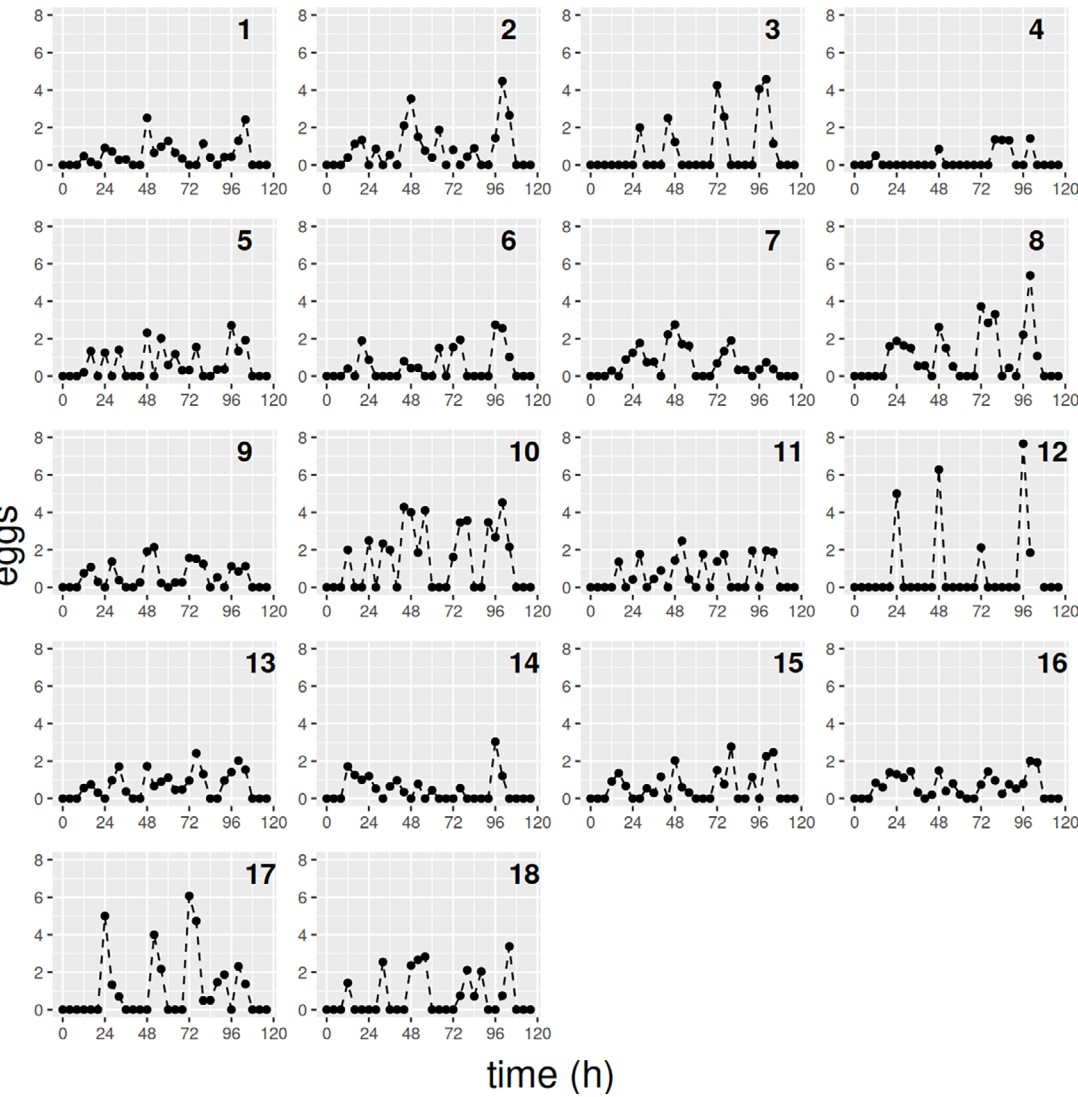

**Appendix 1—figure 15.** Records obtained after applying detrending for the experiment in DD with 18 CantonS female flies described in the text.

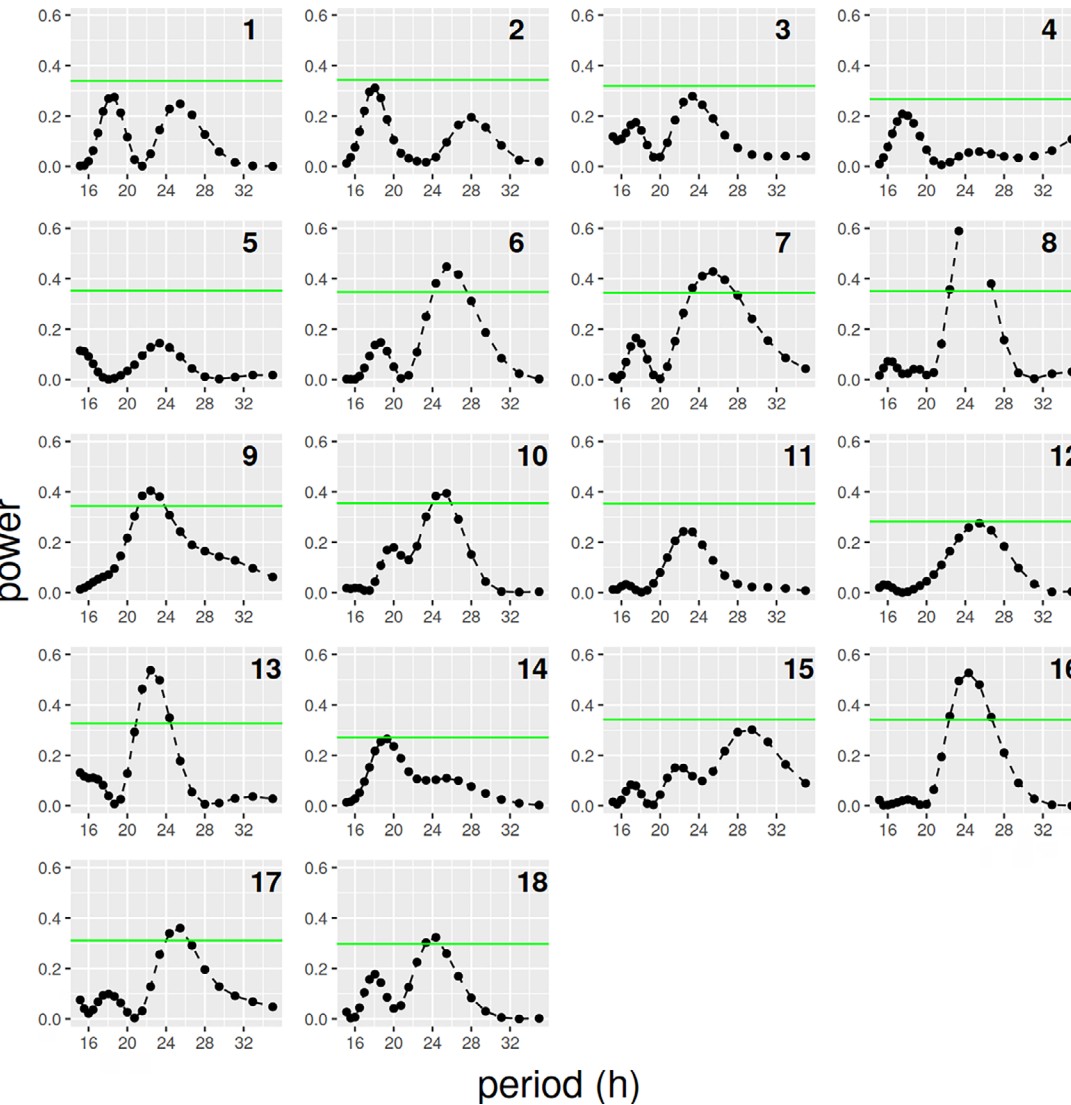

**Appendix 1—figure 16.** Periodograms for the records obtained after applying detrending for the experiment in DD with 18 CantonS female flies described in the text.

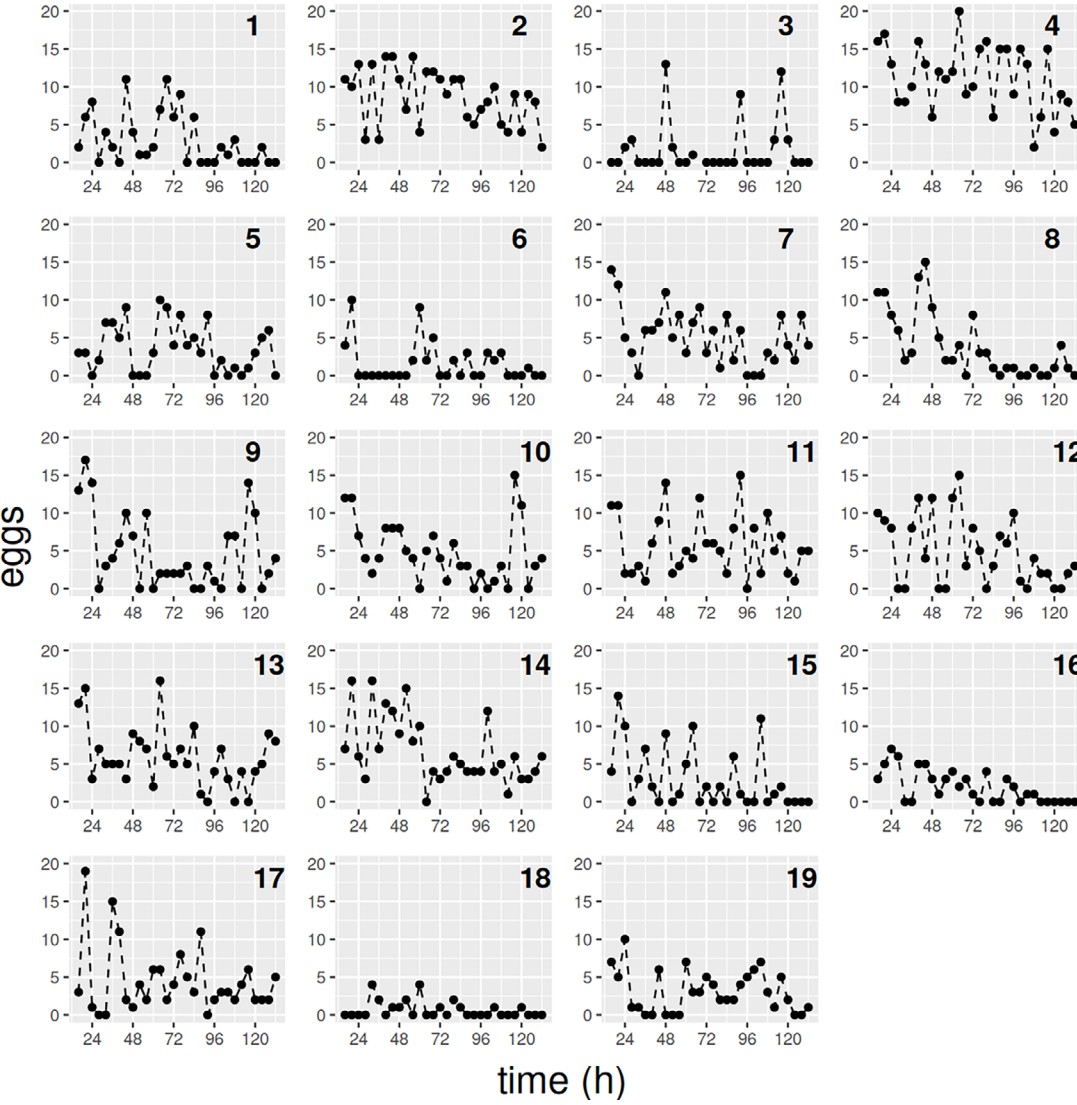

**Appendix 1—figure 17.** Individual egg records of an experiment in DD with 19 per01 female flies.

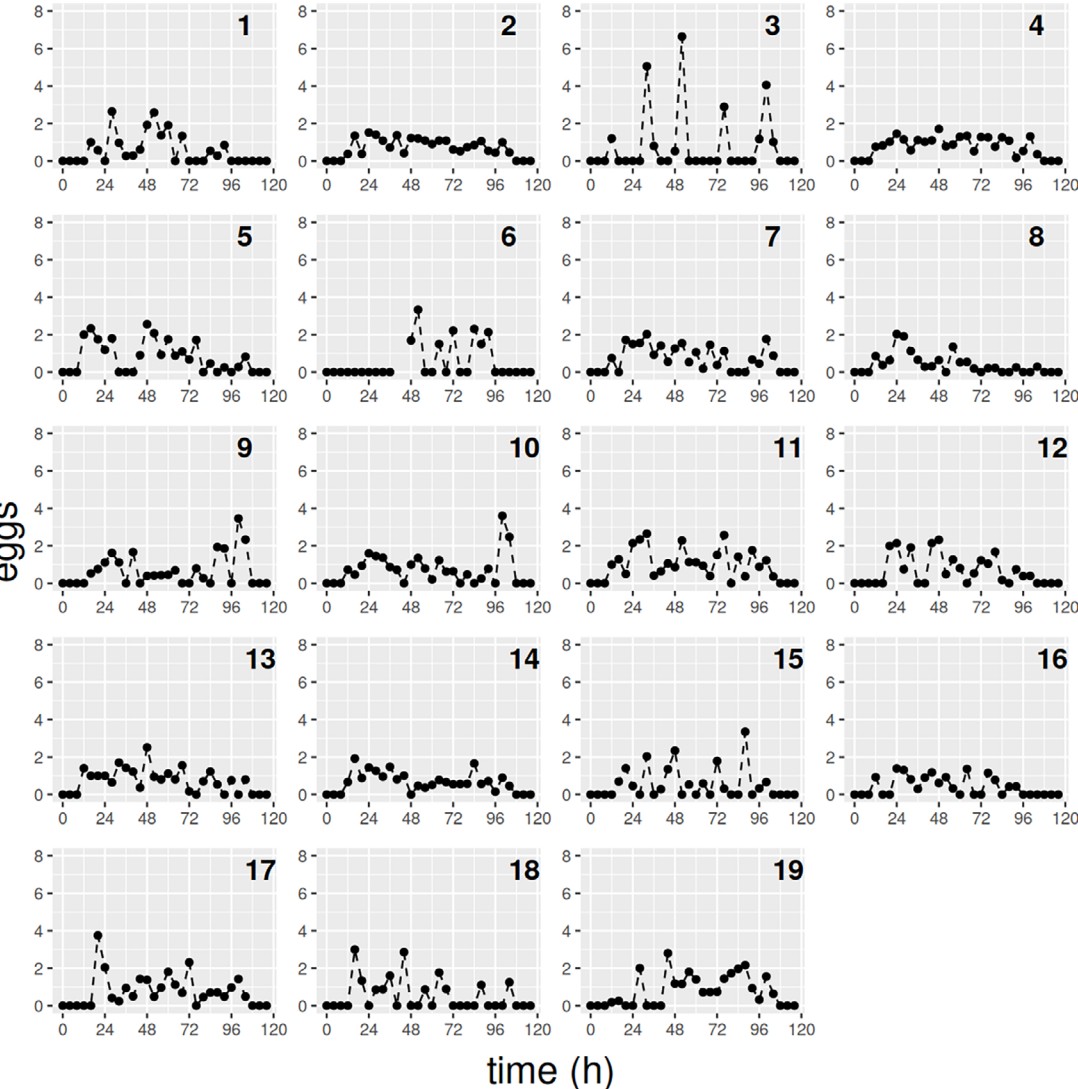

**Appendix 1—figure 18.** Records obtained after applying detrending for an experiment in DD with 19 per01 female flies.

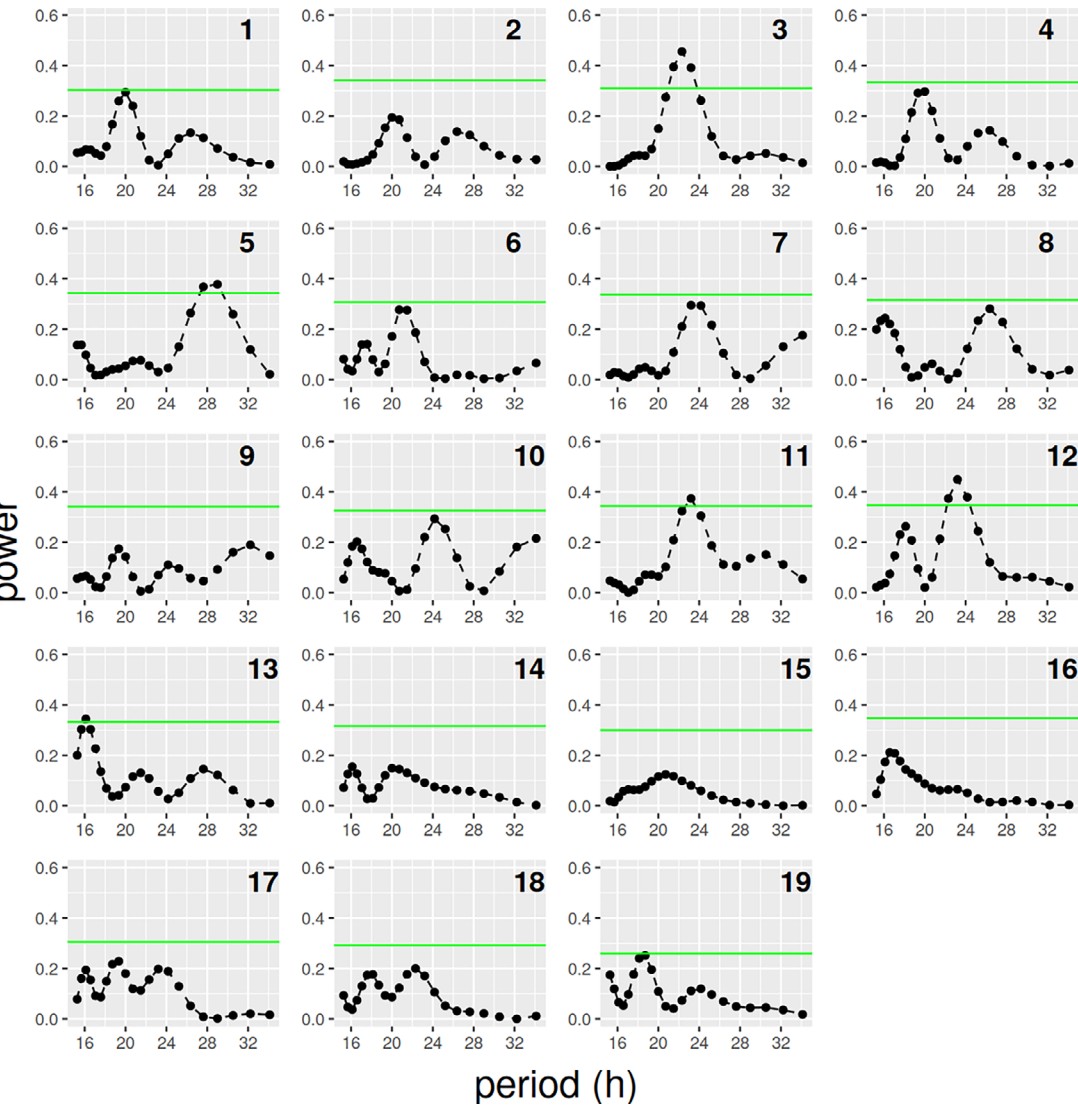

**Appendix 1—figure 19.** Periodograms for the records obtained after applying detrending for an experiment in DD with 19 per01 female flies.

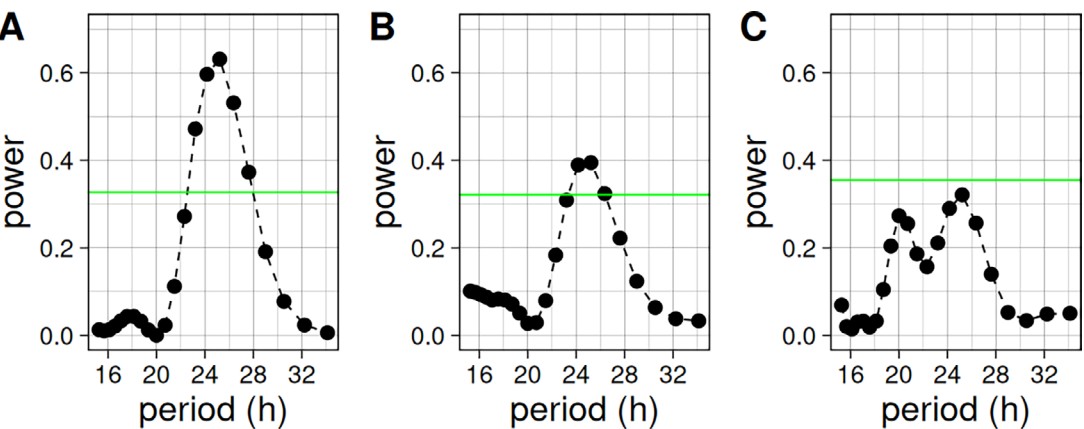

**Appendix 1—figure 20.** Persistence of population rhythmicity with or without individually rhythmic flies.
(**A**) Population periodogram for the experiment with 18 CantonS female flies mentioned in the text. (**B**) Population periodogram for the same experiment as in A, but without eight individually rhythmic flies. (**C**) Population periodogram for the experiment with 19 per01 female flies mentioned in the text.

