## [Editor Report · eLife Assessment]

This **important** study introduces an experimental approach for studying *Drosophila* oviposition rhythms and identifies the subset of circadian clock neurons that mediate the circadian control of oviposition. The authors resolve an inherently noisy rhythm to provide **convincing** evidence by using statistical averaging techniques, which help reduce this noise but at the cost of variation across individual rhythms. This paper will be of interest to anyone interested in insect ovarian physiology, circadian biology, and reproductive fitness.

---

## [Referee Report · Joint Public Review]

Summary

Riva et al. introduce a semi-automatic setup for measuring *Drosophila melanogaster* oviposition rhythms and use it to map the timekeeping function underlying egg laying rhythms to a subset of clock cells. Using a combination of neurogenetic manipulations and referencing the publicly available female hemi-brain connectome dataset, they narrow the critical circuit down to two of the three CRYPTOCHROME expressing lateral-dorsal neurons (CRY[+] LNds). Their findings suggest that different overlapping sets of clock neurons may control different behavioral rhythms in *D. melanogaster*.

This work will be of interest to researchers interested in the circadian regulation of oviposition in *D. melanogaster* (and possibly other insects), a phenomenon which has been left relatively under-explored. The construction of a semi-automated setup which can be made relatively cheaply using available motors and 3D printed molds provides a useful model for obtaining longer records of oviposition activity.

Strengths

The authors use a semi-automated monitoring system to detect circadian egg laying rhythms in spite of inherently noisy data. Using this approach they use a variety of different genetic tools to show that CRY+ LNds play a role in generating the circadian rhythm of oviposition, that PDF-expressing neurons seem to be important for maintaining the circadian period of egg laying, and that period locus function is required for the circadian rhythmicity of oviposition. The authors also point to some potentially interesting connectome data that suggest hypotheses regarding the neuronal circuit linking daily timekeeping to oviposition, which will require further validation in future studies.

Weaknesses:

The major weaknesses of this work result from the noisy nature of the data, and the need to average the individual records of many animals in order to extract significant rhythmicity values. The predicted neural output pathways will require validation in future studies.

---

## [Author Response]

The following is the authors’ response to the previous reviews

**Joint Public Review:**
(1) Problems associated with averaging: The authors intended to focus on the oviposition clock in individual females, however due to the inherent noise in the oviposition rhythm they had to resort to averaging across Lomb-Scargle periodograms generated from individual time-series. They then tested whether the averaged periodogram contains a significant frequency. However, this reduction in noise also reduces the ability to compare differences in power of the rhythm across individuals. Furthermore, this method makes it especially difficult to distinguish the contribution of subsets of the circuit on the proportion of rhythmic flies and the power of the rhythm. In this revised version the authors use two manipulations to disrupt the molecular clock, which could have different success rates based on the type and number of cells targeted. Unfortunately, the type of averaging used prevents the detection of any such effects. It is to be noted that, indeed, individual-level differences in period between the PdfDicerGal4 > perRNAi and UAS-perRNAi lines help the authors to establish that there is a significant reduction in period length when the molecular clock is abolished in PDF cells. These individual measurements are now very helpful in discerning the effect of manipulations carried out on different circadian neural subsets, some of which could have been missed if only averages were considered.

First, it is important to emphasize that we are certainly not "averaging across LombScargle periodograms". As explained in the paper (and at length in the Supplementary Material), what we do is first to detrend each individual time series, then average _all_ the resulting time series (and not only those of rhythmic individuals), and finally take the Lomb-Scargle periodogram of this average series. Nevertheless, we agree with the reviewer in that the use of averages reduces our ability of understanding what happens at the individual level. The problem is that in most cases the presence of noise has made it difficult to draw any meaningful conclusions. One fortunate exception is the one mentioned by the reviewer. Averaging, on the other hand, has allowed us to extract some useful information in those cases.

(2) Sensitivity to sample size: Averaging reduces the effect of random background noise but noise reduction is dependent upon sample size. Comparing genotypes with different sample sizes in addition to varying signal to noise ratios (which might also change with neural manipulations) makes it difficult to estimate how much of the rhythm structure is contributed by a given neuronal subset; thus, whenever possible comparisons should be made between groups that include similar number of flies. This problem is compounded when the averaged periodogram is composed of both rhythmic and weakly rhythmic individuals. For instance, in the main text the reported value of period length of pdfDicerGal4 > perRNAi is 20.74h (see also Fig 2J) but in the Supplementary figure 2S1 this is close to 22h, while the values reported for the control are largely similar (24.35h in Fig 2H versus ~24h in Fig 2S1). A difference of 3.6h between control and experimental flies is much greater than 2h. Which estimate (average versus individual) is more reliable in predicting the behavior of these flies is difficult to determine without further experiments.

In most of the experiments analyzed for this paper the number of flies for control and experimental genotypes are very similar. In the remaining ones, the number of flies for experimental genotypes is roughly twice the number of flies for control genotypes. As mentioned, noise reduction depends on sample size. This implies that, when a genotype is assessed as rhytyhmic the sample size used is evidently large enough. On the other hand, when a genotype is assessed as arrhythmic it is important to know if sample size is large enough. It is for this reason that we have used many more flies for arrhythmic genotypes vs. their control genotypes.

Regarding the period difference between the average of rhythmic individuals, and the population denoised average, notice first that they are not necessarily excactly the same thing, since our population average uses all flies, and the denoising might introduce some variations over the underlying periods (which would be undetectable without the denoising). Also, and more importantly, Fig. 2S1 shows that for the average of the individual periods the error bars are large, and thus statistically, the reported value for the population average falls within the confidence interval for the individual average.

(3) Based on the newly provided data for individual fly periodograms the reader can visually evaluate the rhythmicity associated with each genotype. Such visual inspection did not reveal any clear difference between the proportion of rhythmic individuals between experimental and parental GAL4 and/or UAS controls, except for experiments using per01 mutant animals. This is surprising since if these circuits are controlling the oviposition rhythm, perturbing them should affect most individuals in a similar way.

The problem here is that, given the amount of noise present in this behavior, it is difficult to obtain any reliable information from individual records, since, by its random nature, in a given experiment noise might be disturbing the expected behavior of individuals in very different ways. That is the reason why we have resorted to population averages.

Other commentsDisrupting the clock in the 5th sLNv and 3 Cry+ LNds (and weakly in a small subset of DN1) affected egg-laying. Although the work emphasizes the importance of the LNd, the role of the 5th sLNv's role should be discussed.

As mentioned in the paper, what the experiments show is that the 3 Cry+ LNds and 5th sLNv (usually called E cells) are candidates to be the main drivers of the oviposition rhythm, but the connectomics show that only 2 Cry+ LNds are connected to the oviposition circuit. In order to be more accurate, throughout the corresponding section (now called "The molecular clock in E neurons is necessary for rhythmic egg-laying") of the corrected manuscript we have always referred to the cells marked by the driver as E-cells. In the Discussion, we have added a line commenting that, in the connectome, the 5th sLNv is not connected to any cells of the oviposition circuit.

Minor corrections:

In subsection "Two Cry+ LNd neurons directly oviIN", there was a mistake in the use of "E1" and "E2" (their meanings were interchanged). We have corrected this section, giving the correct definitions. We have also corrected some minor english typos.

**Joint Recommendations for the authors:**
(1) Line 234 'to disrupt the molecular clock in (those) neurons', Please clearly describe the cell types in which MB122B driver works.

We have clarified the cell types in which MB122B driver is expressed (line 236)

(2) Line 235 gen cycle, should be gen'e' cycle

The typo has been corrected

(3) The authors should provide the raw data in repositories as per journal policy of eLife.

The data are now available at the following links:

https://github.com/srisaug/flywork/blob/main/RawData_Rivaetal_eLife2025_Fig4_+> UAS-perRNAi.zip

https://github.com/srisaug/flywork/blob/main/RawData_Rivaetal_eLife2025_Fig4_M 122Bsplit-Gal4>+.zip

https://github.com/srisaug/flywork/blob/main/RawData_Rivaetal_eLife2025_Fig4_MB122Bsplit-Gal4>UAS-perRNAi.zip

https://github.com/srisaug/flywork/blob/main/RawData_Rivaetal_eLife2025_Figures1